# VideoCAD: A Dataset and Model for Learning Long-Horizon 3D CAD UI Interactions from Video

**Brandon Man**[*]   **Ghadi Nehme**[*]   **Md Ferdous Alam**   **Faez Ahmed**
Department of Mechanical Engineering, Massachusetts Institute of Technology
Cambridge, MA 02139, USA
{bm557, ghadi, mfalam, faez}@mit.edu

## Abstract

Computer-Aided Design (CAD) is a time-consuming and complex process, requiring precise, long-horizon user interactions with intricate 3D interfaces. While recent advances in AI-driven user interface (UI) agents show promise, most existing datasets and methods focus on short, low-complexity tasks in mobile or web applications, failing to capture the demands of professional engineering tools. In this work, we introduce VideoCAD, the first attempt to model UI interactions for precision engineering tasks. Specifically, VIDEOCAD is a large-scale synthetic dataset consisting of over 41K annotated video recordings of CAD operations, generated using an automated framework for collecting high-fidelity UI action data from human-made CAD designs. Compared to existing datasets, VIDEOCAD offers an order-of-magnitude increase in complexity for real-world engineering UI tasks, with time horizons up to $20\times$ longer than those in other datasets. We show two important downstream applications of VIDEOCAD: (1) learning UI interactions from professional 3D CAD tools for precision tasks and (2) a visual question-answering (VQA) benchmark designed to evaluate multimodal large language models (LLMs) on spatial reasoning and video understanding. To learn the UI interactions, we propose VIDEOCADFORMER, a state-of-the-art model for learning CAD interactions directly from video, which outperforms existing behavior cloning baselines. Both VIDEOCADFORMER and the VQA benchmark derived from VIDEOCAD reveal key challenges in the current state of video-based UI understanding, including the need for precise action grounding, multi-modal and spatial reasoning, and long-horizon dependencies. The dataset and code are available at: https://github.com/ghadinehme/VideoCAD

## 1 Introduction

Designing most physical products—from cars to airplanes—relies on professional engineering CAD software such as SolidWorks, Autodesk Inventor, and PTC Onshape. These platforms, with their hundreds of toolbars and menu options, pose significant challenges for users due to their complex interfaces, intricate workflows, and the high degree of precision required to create accurate 3D geometries [1, 2]. Unlike typical consumer applications, where tasks are completed through simple User Interface (UI) interactions, CAD operations require structured, multi-step processes involving 3D spatial understanding and parametric modeling. Mastery of these tools often requires years of experience, and automating their use remains a formidable challenge [3]. While recent AI-based approaches aim to automate parts of the CAD workflow, most are limited to text or image modalities [4, 5] and fail to learn from the dynamic, video-based nature of CAD interfaces. The absence of large-scale, annotated datasets capturing such interactions further constrains progress in this area.

---

[*]Equal contribution.

39th Conference on Neural Information Processing Systems (NeurIPS 2025) Track on Datasets and Benchmarks.

Prior research on UI navigation has primarily focused on web and mobile applications, where tasks are short, low in complexity, and require no 3D reasoning. Benchmarks such as MiniWob++ [6] and RICO [7] have provided valuable insights into learning from user interactions, yet they fall short of capturing the complexity of CAD environments—where tasks involve nested dependencies, geometric constraints, and tool-based operations that go far beyond simple button clicks. While behavior cloning from video has shown promise in robotics and gaming, its application to software UI understanding remains largely unexplored due to the lack of large-scale, video-annotated datasets.

To address this gap, we introduce VIDEOCAD, a large-scale synthetic dataset containing over 41K video demonstrations of CAD modeling tasks with fine-grained action annotations. VideoCAD is generated from a public dataset of parametric CAD models, DeepCAD [8], originally authored by human designers in Onshape [9]—a professional browser-based CAD platform (see Appendix A). We map the construction sequences of these CAD models into executable UI actions within Onshape, producing realistic, temporally aligned video–action pairs.

To demonstrate the effectiveness of our dataset, we provide two important downstream applications of VIDEOCAD. First, we develop a transformer-based model, VIDEOCADFORMER, that predicts UI interactions, given a single input image of target CAD model. We benchmark our model against state-of-the-art behavior cloning baselines and show that VIDEOCADFORMER outperforms them by up to 20% in learning from rich user interactions for CAD design, demonstrating superior reasoning over long-horizon UI interaction tasks involving 3D understanding. Second, we also introduce VIDEOCADQA, a synthetically generated multiple-choice VQA dataset derived from VIDEOCAD that includes questions that evaluate 3D reasoning and video understanding in LLMs. Our benchmark helps uncover a critical gap in spatial reasoning of current LLMs in precise engineering tasks.

VIDEOCAD has the potential to serve as a valuable benchmark for advancing research in AI-driven UI navigation, software automation, and CAD generation. By providing an open-source dataset with rich annotations, we aim to facilitate breakthroughs in learning from software demonstrations and bridging the gap between computer vision, reinforcement learning, and CAD modeling. More specifically, our contributions can be summarized as follows:

- **A large-scale, high-fidelity CAD interaction dataset:** VIDEOCAD consists of over 41K annotated videos, capturing diverse CAD workflows with both low-level UI actions and high-level modeling operations.
- **Behavior cloning benchmarks and a state-of-the-art model:** We evaluate multiple baselines on VIDEOCAD and propose VIDEOCADFORMER, a transformer-based architecture that achieves state-of-the-art performance on long-horizon CAD action prediction.
- **A VQA benchmark for 3D and spatiotemporal reasoning:** We present a case study that extends VideoCAD into a VQA benchmark, introducing 1,200 visual questions that probe LLMs' fine-grained 3D reasoning and temporal understanding over CAD videos.

## 2 Related works

Research on automating software understanding spans UI navigation, learning from demonstrations, and 3D CAD generation. We situate our work at the intersection of these areas—linking long-horizon UI modeling with spatially grounded CAD reasoning.

**User Interface Navigation.** Automating user interface navigation has been a long-standing challenge in AI research. Works such as MiniWob++ [10] and Android In The Wild (AITW) [11] introduced datasets for web and mobile UI interaction, allowing agents to learn structured policies for performing simple tasks. However, these datasets focus on relatively simple tasks with short time horizons. More recent works, such as WebShop [12], have extended these capabilities to complex real-world scenarios where an agent must reason over long sequences of actions to achieve its goal.

**UI Agents and Behavior Cloning from Videos.** Learning from demonstrations has been widely applied to UI interaction. Behavioral cloning, where an agent learns from human demonstrations, has been successfully applied in robotic manipulation [13] and game environments [14]. However, its application to GUI-based tasks is still underexplored. Recent studies, such as ActionBert [15] and AssistGUIs [16], explored using large-scale datasets for predicting UI interactions from multimodal inputs. Our work extends these efforts by introducing a dataset specifically tailored for CAD interactions, where the action space is significantly more complex.

**CAD Datasets for Learning-Based Modeling.** Parametric CAD datasets remain scarce compared to mesh or point cloud collections. The ABC dataset offers 1 million B-rep models with precise geometry but lacks procedural data [17]. DeepCAD supplements this by providing 178K models with full construction histories, enabling sequence-based generative modeling [8]. Fusion 360 Gallery contributes 8.6K human-authored CAD programs, capturing sketch-extrude workflows and assembly hierarchies [18]. MFCAD and MFCAD++ focus on machining feature recognition, supplying labeled B-rep models for supervised learning [19, 20]. However, there is a lack of datasets capturing fine-grained UI interactions or the visual and geometric reasoning needed to model 3D CAD workflows. VIDEOCAD addresses this gap by providing video demonstrations of CAD construction, offering temporally grounded UI actions and visual context to support multimodal learning of design behaviors.

**AI-Driven CAD Generation.** Generative modeling of CAD has been an active research area in 3D vision [21, 5, 22, 23]. DeepCAD [8] introduced one of the first deep learning models capable of generating parametric CAD sequences, while subsequent works such as SketchGraphs [24] have provided datasets for learning structured CAD designs. Recent advances in generative AI, such as transformer-based models for sketch generation [25], conditional latent diffusion model for CAD sequence generation from images [26] and direct CAD construction [27], highlight the potential for learning CAD from visual demonstrations. VIDEOCAD complements these efforts by providing an extensive dataset of video demonstrations, enabling new paradigms in learning CAD construction from human-like video demonstrations.

## 3  VIDEOCAD Dataset

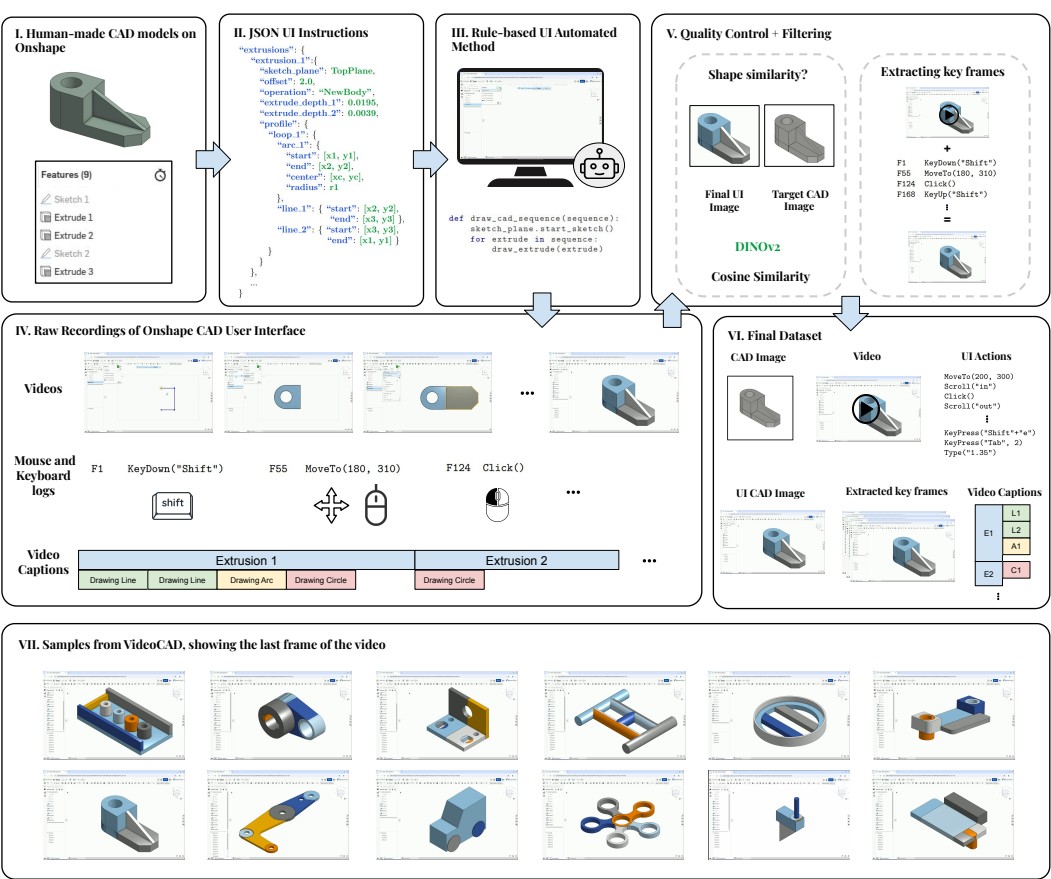

Figure 1: Illustration of the VIDEOCAD dataset pipeline: human-authored CAD sequences are converted into UI instructions and executed via a rule-based automated method to record videos. Quality filtering, keyframe extraction, and action alignment produce structured video-action pairs.

VIDEOCAD is a large-scale dataset comprising 41,005 synthetic videos of 3D CAD model construction, generated from human-authored designs. Each data sample consists of a multiframe video of

an automated agent interacting with the Onshape CAD interface [9], a free, browser-based CAD platform, accompanied by two levels of timestamped action annotations and a ground-truth target rendering in isometric view. The annotations consist of low-level actions and high-level actions. Low-level actions capture UI interactions such as clicking, typing, and mouse movements, with timestamps marking when each action occurred. These low-level actions correspond to the action space discussed in section 4. High-level actions align with CAD modeling operations, recording when primitives such as extrusions and loops are constructed. High-level actions correspond to the CAD primitives found in DeepCAD. More specifically, VIDEOCAD consists of the following, $\mathcal{D} = \{(\mathbf{X}_i, \mathbf{I}_i, \mathbf{a}_i)\}_{i=1}^N$ where $\mathbf{X}_i \in \mathbb{R}^{t \times H \times W \times C}$ is the video frame, $\mathbf{I}_i \in \mathbb{R}^{H' \times W' \times C'}$ is the corresponding image of the target shape, and $\mathbf{a}_i \in \mathbb{R}^{t \times d}$ is the corresponding action vector that represents the UI actions taken to create the CAD with the desired shape.

## 3.1 Dataset Generation Pipeline

**CAD Construction Sequence.** To create the VIDEOCAD dataset, we leverage the DeepCAD sequence representation [8], which models CAD construction as a sequence of sketches and extrusions. Each 2D sketch consists of one or more closed loops formed by geometric primitives—lines, arcs, and circles—parameterized by their spatial properties (e.g., start/end points, midpoints, radii). These sketches are extruded using parameters such as orientation (Euler angles $\theta, \phi, \gamma$), 3D offsets $(p_x, p_y, p_z)$, scale $s$, bidirectional depths $(e_1, e_2)$, and Boolean operations $\beta$ (e.g., join, cut). This design framework is visualized in Figure 2.

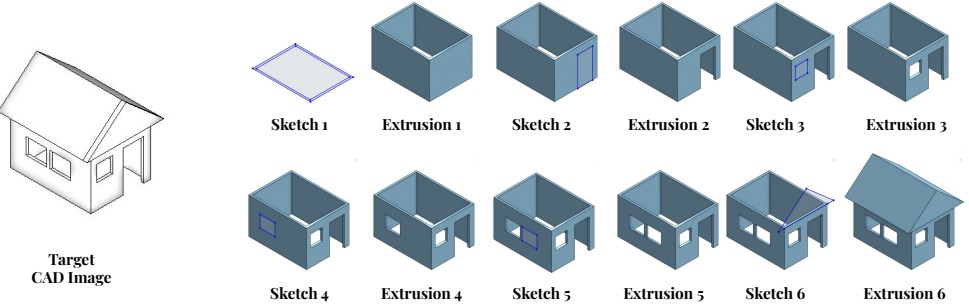

Figure 2: **Example of intermediate modeling stages in VIDEOCAD.** A sequence of snapshots illustrating the progressive construction of a CAD model through successive sketching and extrusion operations.

**From Human-Created CAD Models to UI Instructions.** We systematically convert these sequences of human-made CAD models into executable sequences of UI actions compatible with Onshape's interface. To automate CAD modeling within Onshape, we developed a hybrid UI interaction framework driven by a rule-based automated method. Each CAD model is built incrementally by defining sketch planes, either default or custom offset planes, followed by drawing geometric primitives (lines, arcs, circles), and performing extrusion operations with specified parameters. Further details on the translation from DeepCAD sequences to CAD UI actions are described in Appendix L. A complementary discussion on the difference between high-level *CAD sequences* and low-level *UI sequences* can be found in Appendix O

**Rule-based UI Automated Method.** We execute the UI instructions within the Onshape interface using a hybrid rule-based method. The mapping from high-level UI instructions to low-level UI actions is detailed in Appendix M. The method combines Selenium for Document Object Model (DOM)-level automation and PyAutoGUI for pixel-level input, enabling end-to-end sketching and extrusion without requiring Onshape's internal API (Appendix K). To enhance realism and downstream learnability, we inject human-like heuristics, including randomized delays, surface point sampling, and zooming on small features—simulating natural user behavior in long-horizon CAD modeling tasks [28]. During execution, we record the full screen to capture the visual feedback of the automated construction of the CAD model, and we log both the low-level UI actions and the corresponding high-level CAD commands as structured video captions.

**Quality Control and Filtering.** To ensure high-quality reconstruction, we render the final CAD model from an isometric view and compare it with the human-authored reference using DINOv2

vision embeddings. A cosine similarity threshold is applied to automatically discard inaccurate reproductions. We manually verified that this metric correlates with geometric correctness (see Appendix N). After filtering, keyframes are extracted for each UI action based on the logged frame index, yielding a sequence of temporally aligned image–action pairs.

## 3.2 Dataset Composition and Statistics

We focus exclusively on multi-extrusion sequences from the DeepCAD dataset to create VIDEOCAD. Multi-extrusion sequences involve substantially longer action sequences (Figure 12) and multi-surface operations–making them more challenging for learning-based models. Each episode includes: 1) A rendered target image of shape $3 \times 224 \times 224$, in isometric view, 2) A full-resolution video at 1600×1000, recorded at 60 FPS, 3) A sequence of action tuples aligned with the video timeline. Figure 3 shows the action commands distribution. Appendix J provides additional plots on dataset statistics and representative CAD examples across complexity levels.

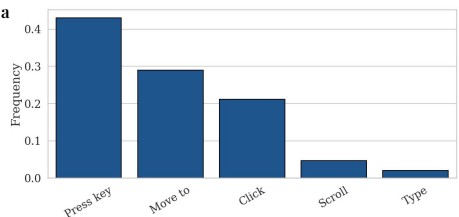 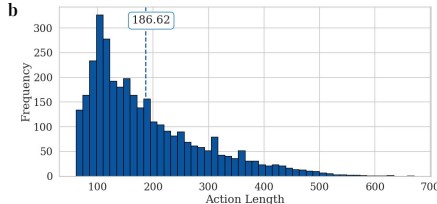

Figure 3: Statistical distributions of CAD UI actions and UI sequence lengths. **a.** Action command frequencies. **b.** UI sequence length frequencies.

## 3.3 Benchmarking

Engineering UI is often more complex than traditional UI due to the necessity of many operations for precision control of the user. This can be seen in CAD software due to the complexity of the UI needed for precise spatial reasoning. To show this complexity, we benchmark VIDEOCAD extensively against other existing UI-agent datasets from the literature in Table 9 (Appendix I).

| Environment | # Samples | Time Horizon | 3D Reasoning | Precise Elements | Avg. # Elements |
|---|---|---|---|---|---|
| OSWorld [29] | 369 | 15* | ✗ | ✓ | – |
| Mind2Web [30] | 2,350 | 7.3 | ✗ | ✗ | 1,135 |
| WebShop [12] | 12,000 | 11.3 | ✗ | ✗ | 38 |
| WebLinx [31] | 2,337 | 43 | ✗ | ✗ | 1,849 |
| AITW [11] | **715,142** | 6.5 | ✗ | ✗ | – |
| MMINA [32] | 1,050 | 12.9 | ✗ | ✓ | 601 |
| MetaGUI [33] | 1,125 | – | ✗ | ✗ | 79 |
| MoTIF [34] | 4,707 | 4.4 | ✗ | ✗ | 188 |
| GUI-WORLD [35] | 12,379 | 10.97 | ✓ | ✓ | – |
| **VIDEOCAD** | 41,005 | **186** | ✓ | ✓ | **6,740** |

Table 1: Comparison of multi-environment benchmarks for GUI interaction. * The max is used instead of the average as the average is not reported.

We compare dataset complexity across five metrics: **# Samples** – total number of samples; **Time Horizon** – number of UI actions per task; **Requires 3D Reasoning** – whether tasks involve manipulating 3D coordinates; **Precise Element** – whether agents must act via *xy* coordinates (e.g., canvases) rather than DOM selectors, requiring spatial and visual reasoning; and **Average UI Elements** – mean number of HTML elements per interface (reported only for datasets with HTML tree access).

As shown in Table 1, VIDEOCAD stands out across multiple dimensions. It features the longest time horizon—4× that of the next closest dataset (WebLinx)—and is the second-largest dataset after Android in the Wild, with over 50× more samples than the median (812). It is one of only two datasets requiring 3D spatial reasoning and pixel-level *xy* interactions, challenging AI models to operate beyond text-based commands. For datasets with DOM access, VIDEOCAD also contains 6× more UI elements than the average web page in Mind2Web.

# 4   VideoCADFormer: An Autoregressive Transformer to Predict CAD Actions

**Commercial CAD Software as an Environment.**   We model CAD construction as a sequential decision process, where an agent observes UI frames and predicts low-level actions to recreate a target 3D model. Each trajectory in VIDEOCAD is a sequence $\tau = \{(\mathbf{I}, \mathbf{o}_t, \mathbf{a}_t)\}_{t=0}^T$, where $\mathbf{I}$ is a fixed image of the target shape, $\mathbf{o}_t$ is the UI frame at timestep $t$, and $\mathbf{a}_t$ is the expert action. The agent learns a policy $\pi_\theta(\mathbf{a}_t \mid \mathbf{o}_{t-k:t}, \mathbf{I})$ via behavior cloning, trained to minimize a supervised loss over expert demonstrations $\mathcal{L}_{\text{BC}} = \mathbb{E}_{\tau \sim \mathcal{D}} \sum_t \ell\left(\pi_\theta(\mathbf{o}_{t-k:t}, \mathbf{I}), \mathbf{a}_t\right)$, where $\ell$ is a structured prediction loss over commands and parameters. Our setup can be extended to reinforcement learning with rewards based on geometric similarity (see Appendix C for Chamfer Distance reward details).

**Observation Space.**   At each timestep $t$, the model observes a grayscale UI image $\mathbf{o}_t \in \mathbb{R}^{224 \times 224 \times 1}$ representing the current design state, and a target CAD image $\mathbf{I} \in \mathbb{R}^{224 \times 224 \times 1}$, which remains fixed throughout the episode. Together, these visual inputs provide both local progress and global goal context, enabling the model to ground its predictions in both current and intended geometry.

**Action Space.**   To interact with the CAD interface, the agent issues low-level UI commands. Each action $\mathbf{a}_t$ is a structured tuple: $\mathbf{a}_t = (c_t, p_t^1, \dots, p_t^{d_t})$, $c_t$ is the command type index, $d_t$ is the number of parameters for command $c_t$, $p_t^i$ is the $i$-th parameter for command $c_t$. Each action is a fixed-length vector: $\mathbf{a}_t = (c_t, x_t, y_t, k_t, n_t, s_t, v_t)$, where each field corresponds to the parameter(s) used by one or more commands. Table 2 is the set of low-level UI commands and their associated parameters. Unused fields are padded with $-1$. All parameter values are discretized into 1000 classes, enabling the action prediction task to be cast as a multi-class classification problem. At the end of each CAD video, we set the part in isometric view; this action serves as our end-of-sequence command.

| Command | Description | Parameters |
|---|---|---|
| MoveTo | Move pointer to screen coordinate | $x_t, y_t$ : pointer location |
| PressKey | Press keyboard key(s) | $k_t$ : key index,   $n_t$ : press count |
| Scroll | Scroll to zoom or pan | $s_t$ : scroll amount |
| Type | Enter numerical value | $v_t$ : typed value |
| Click | Left mouse click | $\emptyset$ |

Table 2: Structured action representation in VIDEOCAD. Each command type is mapped to a consistent 7D vector used for classification. Unused fields are set to $-1$.

## 4.1   Model Architecture

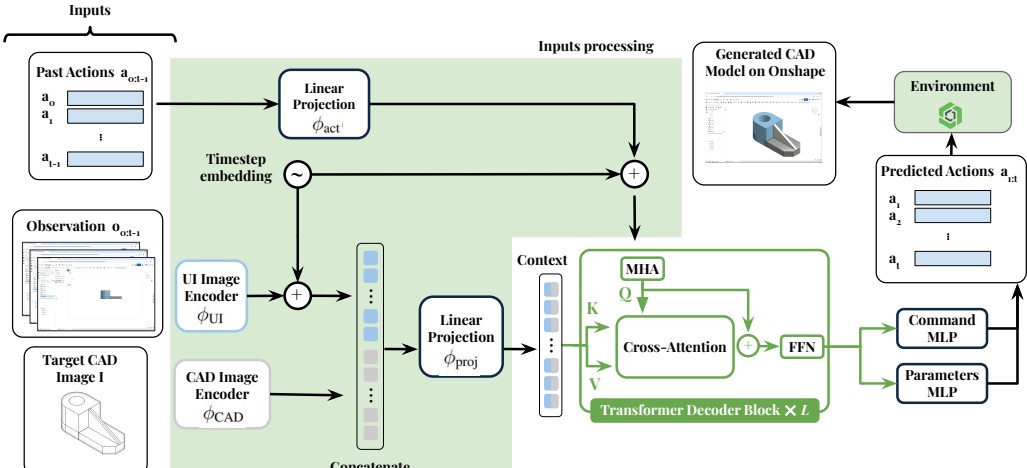

Figure 4: Overview of VIDEOCADFORMER for CAD UI action prediction. The model encodes the target image and past UI frames via ViT, fuses them with projected past actions using a cross-attention decoder, and predicts the next action to iteratively build the CAD model in Onshape.

We propose VIDEOCADFORMER, an autoregressive transformer tailored for CAD UI modeling (Figure 4). The model predicts low-level actions conditioned on a target CAD image (sketch, kernel-rendered, or realistic), past UI-rendered frames, and prior actions. Visual inputs are processed by separate ViT encoders and fused via linear projection with timestep embeddings, capturing static goals and canvas evolution. Action tokens are embedded and passed to a causal transformer decoder with two key inductive biases: a causal mask to enforce autoregressive prediction, and a window mask to focus attention on recent context—reflecting the short-term dependency of UI commands. The decoder outputs command and parameter logits through two heads, with command-dependent masking to suppress irrelevant parameters. Full training details and ablation studies are in Appendix B.

**Input Representation.** At each timestep $t$, the model receives: a target CAD image $I \in \mathbb{R}^{H \times W \times 1}$, a sequence of past UI-rendered frames $o_{t-k:t-1} \in \mathbb{R}^{k \times H \times W \times 1}$, a sequence of previous actions $a_{0:t-1} \in \mathbb{R}^{k \times d}$, and timestep embeddings $\phi_{\text{time}}(t) \in \mathbb{R}^{h}$.

**Visual Encoding.** Each input image (target CAD and UI-rendered frames) is processed through a ViT encoder followed by a linear projection into a hidden dimension $h$. We define:

$$v^I = \phi_{\text{CAD}}(I) \in \mathbb{R}^h, \qquad v_t^o = \phi_{\text{UI}}(o_t) + \phi_{\text{time}}(t) \in \mathbb{R}^h$$

The sequence of UI frame embeddings is concatenated with the static CAD embedding and projected:

$$z_t^{\text{image}} = \phi_{\text{proj}}([v^I; v_t^o]) \in \mathbb{R}^h.$$

**Action and Timestep Embeddings.** Each previous action $a_\tau \in \mathbb{R}^d$ is projected to the hidden space:

$$z_\tau^{\text{act}} = \phi_{\text{act}}(a_\tau) + \phi_{\text{time}}(\tau), \quad z_\tau^{\text{act}} \in \mathbb{R}^h.$$

All inputs are activated using $\tanh$ and passed to the transformer decoder.

**Transformer Decoder.** We use an $L$-layer causal transformer decoder with hidden size $h$ and $n$ attention heads. The decoder operates over a sequence of action embeddings $Z^{\text{act}} \in \mathbb{R}^{T \times h}$ (target) and visual memory $Z^{\text{image}} \in \mathbb{R}^{T \times h}$ (source):

$$H_t = \text{TransformerDecoder}(Z^{\text{act}}, Z^{\text{image}}, \mathbf{M}^{\text{causal}}, \mathbf{M}^{\text{window}}).$$

We use two attention masks: a **causal mask** $\mathbf{M}^{\text{causal}} \in \mathbb{R}^{T \times T}$ is an upper-triangular matrix with $-\infty$ in positions where future tokens should be masked and $0$ elsewhere

**Action Prediction.** The hidden states $H_t \in \mathbb{R}^{T \times h}$ are decoded into actions using two heads:

$$\hat{c}_t = \text{softmax}(W_c H_t + b_c), \quad \hat{c}_t \in \mathbb{R}^5, \qquad \hat{p}_t = \text{softmax}(W_p H_t + b_p), \quad \hat{p}_t \in \mathbb{R}^{6 \times 1000}.$$

Command-dependent masks $M_{\hat{c}_t} \in \{0, 1\}^6$ are applied to suppress unused parameter outputs. Invalid values are set to $-1$.

## 4.2 Evaluation Metrics

**Command and Parameter Accuracy.** We report the classification accuracy of predicted commands $\hat{c}_t$ and parameters $\hat{p}_t$ across the entire test set. Command accuracy measures the fraction of correctly predicted command types. Parameter accuracy is computed per action, conditioned on correct command prediction.

**Offline Closed-Loop Execution Performance.** We evaluate the stability of models under full-sequence autoregressive rollouts by computing the percentage of perfectly predicted actions—defined as exact matches with the ground truth—across all sequences in the test set. We report the mean, minimum, and maximum values of this percentage per method and breakdowns by sequence length. Sequences are categorized into short (0–120), medium (120–200), and long (200+) bins based on test set percentiles.

**Geometric Fidelity.** We evaluate geometric accuracy by executing model-predicted actions in Onshape and rendering the resulting CAD models. On 200 test sequences, we compute mean bidirectional Chamfer Distance (CD) after PCA alignment (Appendix C). A sequence is considered successful if CD $< 0.02$, a threshold chosen empirically based on human evaluation, where shapes below this value are visually indistinguishable (Appendix C). Invalid sequences (i.e., those failing to mesh) are also reported.

# 5   Results

**Accuracy and Perfect Sequences.** We use Video Pre-training (VPT) [14] (a leading method in offline behavior cloning for Minecraft), Pix2Act [36] and Pearce et. al. [37] as baselines. As shown in Table 6, VIDEOCADFORMER achieves the highest command and parameter accuracy across all evaluated models, outperforming VPT, Pix2Act, and Pearce et al. It also leads across all metrics of perfect action sequence prediction. The model shows consistent improvements over VPT across all levels of task difficulty. These results highlight VIDEOCADFORMER's robustness in executing complete CAD sequences without error, particularly in long-horizon settings where small mistakes compound. The increasing performance gap with task difficulty indicates better generalization under growing complexity.

**Geometric Fidelity.** Table 4 evaluates the final CAD quality by executing the predicted sequences in Onshape. VIDEOCADFORMER outperforms VPT in both short and long sequences in terms of Chamfer-based success rate, while VPT performs slightly better on medium sequences. Overall, VIDEOCADFORMER achieves a lower Chamfer distance and a lower proportion of invalid CAD models. These results suggest that stronger sequence-level prediction directly translates to more accurate 3D model reconstruction.

Table 3: Evaluation metrics across task difficulty levels.

| | $\mu_{\textbf{cmd}}$ (%) | $\mu_{\textbf{param}}$ (%) | Perfectly Predicted Actions (%) | | | | | |
| --- | --- | --- | --- | --- | --- | --- | --- | --- |
| | | | **Mean** | **Max** | **Min** | **Short** | **Medium** | **Long** |
| Pix2Act [36] | 20.44 | 2.61 | 2.84 | 22.03 | 0.00 | 2.28 | 2.63 | 3.60 |
| Pearce et al. [37] | 42.60 | 0.55 | 0.68 | 10.80 | 0.00 | 0.83 | 0.69 | 0.51 |
| VPT [14] | 96.25 | 78.72 | 83.81 | **100.00** | 43.59 | 88.51 | 82.77 | 80.12 |
| VIDEOCADFORMER | **98.08** | **82.35** | **87.54** | **100.00** | **65.67** | **90.08** | **87.08** | **85.46** |

Table 4: Performance by sequence length on Chamfer success rate ($<0.02 \uparrow$), mean Chamfer distance ($\downarrow$), and invalid sample rate ($\downarrow$). Overall metrics are averaged across categories.

| Method | Success Rate (%) $\uparrow$ | | | | Mean CD $\downarrow$ | | | | Invalid (%) $\downarrow$ |
| --- | --- | --- | --- | --- | --- | --- | --- | --- | --- |
| | Short | Medium | Long | Mean | Short | Medium | Long | Mean | |
| Human Expert | **85.0** | **96.7** | **82.8** | **88.2** | **0.0097** | **0.0067** | **0.0112** | **0.0092** | **0.0** |
| Random | 2.5 | 0.0 | 0.0 | 0.8 | 0.1038 | 0.1075 | 0.0972 | 0.1028 | – |
| VPT | 43.9 | 36.4 | 5.9 | 28.5 | 0.0260 | 0.0290 | 0.0856 | 0.0473 | 39.5 |
| VIDEOCADFORMER | **60.6** | 37.9 | 25.0 | 41.0 | 0.0238 | 0.0286 | 0.0592 | 0.0374 | 26.0 |

## 5.1   Goal-Driven CAD Generation and Autocompletion

**Shape Generation from Scratch.** Conditioned only on a target CAD image, our model can generate full action sequences that construct the corresponding geometry from an empty canvas. As shown in Figure 5, the predicted low-level UI actions reliably recreate complex multi-step CAD models. This demonstrates the model's capacity to plan and execute long-horizon sequences that align with spatial goals purely from visual context. Furthermore, we train models to take any of the three image types—sketch, kernel-rendered, or realistic—as input representations of the target CAD geometry (see Appendix B for details).

**CAD Model Autocompletion.** Beyond generation from scratch, our model can also autocomplete a partially built CAD model when conditioned on both the current intermediate state and a target final image. Given a prefix of UI actions (e.g., an initial sketch and extrusion), the model predicts the remaining sequence needed to match the final design. As shown in Figure 5, this enables intelligent continuation of in-progress designs and assistive AI behavior in iterative modeling workflows.

**Failure Cases and Limitations.** Model failures are primarily due to inaccurate $(x, y)$ predictions that cause open or slightly distorted sketch loops, preventing valid extrusions. Misclassification between lines and arcs also occurs, especially when curvature is ambiguous. These errors (Figure 10) are often

minor and correctable, but they highlight the limitations of image-only supervision and suggest the need for topological constraints or interactive fine-tuning. More details in Appendix F.

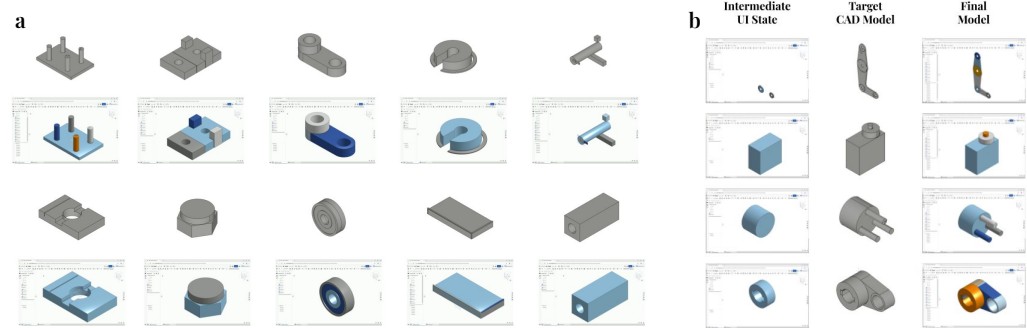

Figure 5: Predicted CAD models from VIDEOCADFORMER, conditioned on (a) a target image for generation from scratch, and (b) a partial UI state for autocompletion.

## 5.2 Case Study: Evaluating LLMs on 3D Reasoning and CAD Understanding

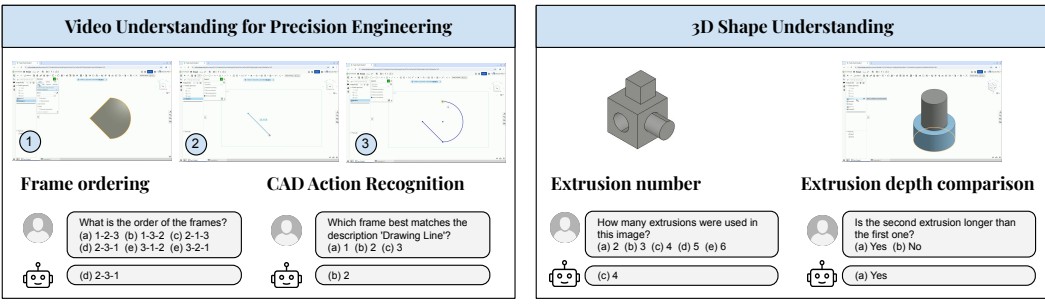

Figure 6: Example questions from the VIDEOCAD VQA benchmark.

| Model | Action Recognition | Frame Ordering | Extrusion Number | Extrusion Comparison |
|---|---|---|---|---|
| gpt-4.1-2025-04-14 [38] | **84.1** | 36.0 | **47.0** | 73.5 |
| claude-3-7-sonnet-20250219[39] | 80.1 | 23.0 | 37.5 | **80.0** |
| qwen2.5-vl-72b-instruct[40] | 78.6 | 32.5 | **47.0** | 62.0 |
| o3-2025-04-16 [41] | 75.1 | **80.0** | 45.0 | 71.5 |
| gemini-2.5-pro-preview-05-06 [42] | 82.6 | 73.2 | 38.0 | 71.0 |
| random | 40.3 | 17.4 | 21.4 | 49.1 |

Table 5: Comparison of models on CAD reasoning tasks. Metrics are in %.

**LLMs Struggle with 3D Reasoning and CAD Video Understanding.** A key capability for AI models to create 3D CAD designs is to excel in 3D reasoning. To evaluate the limits of modern multimodal LLMs in spatially grounded engineering domains, we construct a VQA benchmark, VIDEOCAD VQA, which is derived from VIDEOCAD. This benchmark contains multiple-choice multimodal questions that probe fine-grained understanding of both temporal video sequences and 3D geometric properties—tasks essential for CAD modeling. It includes four task types: action recognition, frame ordering, extrusion counting, and depth comparison. Questions are generated automatically from ground-truth UI logs and CAD geometry (Appendix H).

We perform zero-shot evaluation on VIDEOCAD VQA by feeding a template followed by the prompt to each LLM. If there is no valid answer in the model's response, we perform random selection as a

remedy. We evaluate the model three times for each question. Table 5 shows results across multimodal LLMs. Despite their strong performance on general VQA benchmarks, current models underperform across most categories. For instance, GPT-4.1 achieves only $47\%$ accuracy on extrusion counting and $18\%$ on depth estimation, which highlights persistent challenges in grounded visual reasoning and geometry-aware token alignment. These results underscore a critical gap in LLMs' ability to interpret complex video-based modeling workflows. Importantly, this VQA benchmark serves as a case study showcasing how VIDEOCAD enables rigorous evaluation of spatial and procedural understanding in LLMs. Many additional multimodal tasks—such as symmetry detection, subpart recognition, or parameter estimation—can be derived from the same dataset and are discussed in Appendix H.

**LLMs Fail as UI Agents for Precision CAD Tasks.** We also benchmark LLMs as UI agents in complex engineering software. Using BrowserGym [43], we prompt models to perform CAD construction tasks within Onshape based only on a target screenshot. Each agent is given 200 steps and must produce *xy* coordinates corresponding to UI actions (e.g., drawing, extrusion). Tasks are drawn from the 10 shortest sequences in VIDEOCAD, with early termination if no meaningful canvas modification occurs. All LLMs—including GPT-4.1, Claude-3.7, and Gemini-2.5—fail to complete a full CAD construction (Appendix ). We anticipate that this happens due to the need for pixel-level precision, long-horizon spatial planning, and consistent 3D reasoning capabilities for CAD modeling unlike web tasks (e.g., spreadsheets or browser automation [44]).

## 6 Limitations and Future Work

While VIDEOCAD offers a high-fidelity benchmark for CAD UI modeling, it has several limitations. All trajectories are synthetically generated by a rule-based bot, and despite human-inspired heuristics, they lack natural variability in timing, errors, and strategy. The dataset focuses solely on sketch-extrude operations, omitting operations like fillets, sweeps, and lofts. Interactions are limited to a single platform—Onshape—raising concerns about generalization to other CAD software. Finally, our current end-to-end benchmarks are only validated on a small subset due to the high cost of CAD rendering and geometric comparison. To address these gaps, future work will: **(1)** incorporate human demonstration data (e.g., from YouTube CAD tutorials), **(2)** extend coverage to assemblies and more advanced CAD features, **(3)** support additional CAD platforms (e.g., Fusion 360, FreeCAD), **(4)** collect multiple trajectories per CAD target to capture variation across users, **(5)** introduce extrusion-by-extrusion text prompts to enable natural interaction between users and AI agents in CAD softwares.

## 7 Conclusion

In this work, we introduced VIDEOCAD, a dataset for evaluating agents on utilizing complex computer software. Our dataset is significantly more complex than other datasets in number of action sequences and number of elements, as well as requiring agents to reason about 3D geometric spaces. VIDEOCAD extends beyond imitation learning and 3D reasoning to serve as a versatile benchmark across machine learning subfields. Its long, fully-observed action sequences support reinforcement learning and planning methods. The alignment between video, symbolic actions, and 3D outputs enables large-scale multimodal pre-training, which can be fine-tuned for tasks like action prediction, video segmentation, and shape retrieval. Its structured, geometry-changing workflows also make it valuable for research in computer vision (e.g., goal inference), HCI (e.g., tutorial generation), robotics (e.g., skill learning), and NLP (e.g., grounding language in actions). VIDEOCAD offers a foundation for developing generalist agents that can perceive, act, and reason in complex software environments.

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

## Table of Contents for Appendices

# A CAD Software

We use Onshape [9], a cloud-native CAD platform, as the environment for our task of learning CAD UI for creating and editing 3D models of mechanical parts and assemblies. Note that unlike traditional desktop-based CAD software, Onshape runs entirely in the browser and supports real-time collaboration, version control (similar to Git), and parametric design. We chose Onshape as it is free, platform agnostic (a majority of CAD software run on Windows), and highly accessible via the browser. In addition, Onshape contains most of the CAD operations used in standard commercial CAD software. We choose to use Google Chrome as the browser of choice as it is available on all main operating systems.

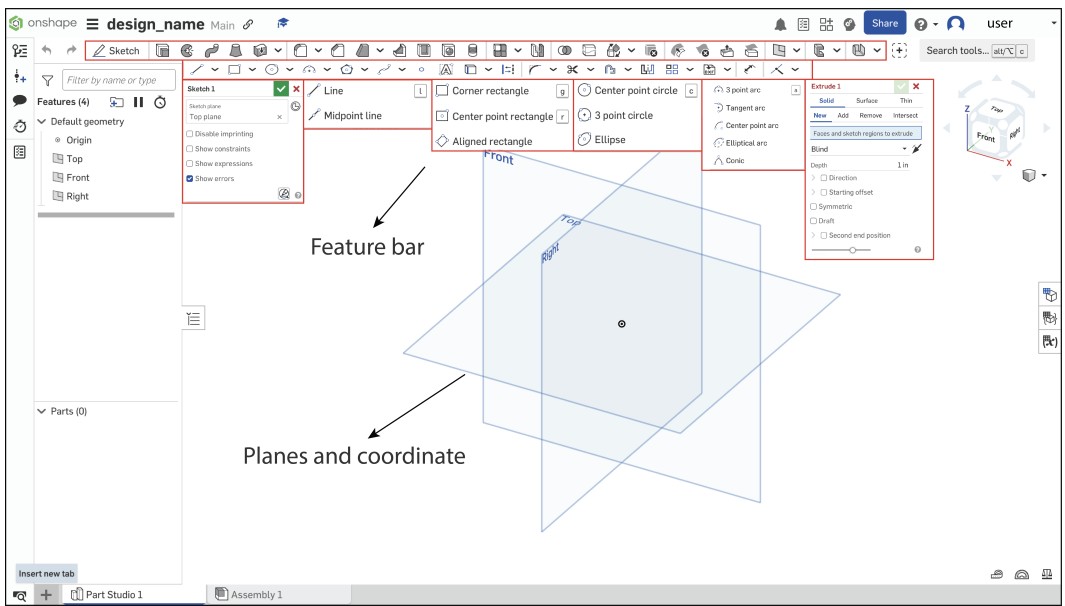

Figure 7: Onshape user interface

# B Model Architecture and Ablation Studies

**Input Representation.** At each timestep $t$, the model receives: a target CAD image $I \in \mathbb{R}^{H \times W \times 1}$, a sequence of past UI-rendered frames $o_{t-k:t-1} \in \mathbb{R}^{k \times H \times W \times 1}$, a sequence of previous actions $a_{0:t-1} \in \mathbb{R}^{k \times d}$, and timestep embeddings $\phi_{\text{time}}(t) \in \mathbb{R}^h$.

**Visual Encoding.** Each input image (target CAD and UI-rendered frames) is processed through a ViT encoder followed by a linear projection into a hidden dimension $h$. We define:

$$v^I = \phi_{\text{CAD}}(I) \in \mathbb{R}^h, \qquad v_t^o = \phi_{\text{UI}}(o_t) + \phi_{\text{time}}(t) \in \mathbb{R}^h$$

The sequence of UI frame embeddings is concatenated with the static CAD embedding and projected:

$$z_t^{\text{image}} = \phi_{\text{proj}}([v^I; v_t^o]) \in \mathbb{R}^h.$$

**Action and Timestep Embeddings.** Each previous action $a_\tau \in \mathbb{R}^d$ is projected to the hidden space:

$$z_\tau^{\text{act}} = \phi_{\text{act}}(a_\tau) + \phi_{\text{time}}(\tau), \quad z_\tau^{\text{act}} \in \mathbb{R}^h.$$

All inputs are activated using $\tanh$ and passed to the transformer decoder.

**Transformer Decoder.** We use an $L$-layer causal transformer decoder with hidden size $h$ and $n$ attention heads. The decoder operates over a sequence of action embeddings $Z^{\text{act}} \in \mathbb{R}^{T \times h}$ (target) and visual memory $Z^{\text{image}} \in \mathbb{R}^{T \times h}$ (source):

$$H_t = \text{TransformerDecoder}(Z^{\text{act}}, Z^{\text{image}}, \mathbf{M}^{\text{causal}}, \mathbf{M}^{\text{window}}).$$

We use two attention masks: a **causal mask** $\mathbf{M}^{\text{causal}} \in \mathbb{R}^{T \times T}$ is an upper-triangular matrix with $-\infty$ in positions where future tokens should be masked and $0$ elsewhere

**Action Prediction.** The hidden states $H_t \in \mathbb{R}^{T \times h}$ are decoded into actions using two heads:

$$\hat{c}_t = \text{softmax}(W_c H_t + b_c), \quad \hat{c}_t \in \mathbb{R}^5, \qquad \hat{p}_t = \text{softmax}(W_p H_t + b_p), \quad \hat{p}_t \in \mathbb{R}^{6 \times 1000}.$$

Command-dependent masks $M_{\hat{c}_t} \in \{0,1\}^6$ are applied to suppress unused parameter outputs. Invalid values are set to $-1$.

For VPT, we feed the UI Images into the CNN layer proposed by the paper, and then concatenate the output representation, the CAD image representation and the past actions and feed them into a feed forward network to get the command and parameters. For Pix2Act, we concatenate the CAD image and the past 10 images into a single image and feed it into the model, we then take the final hidden representation from the decoder and feed them into two linear layers that predict command and action parameters. For Pearce et. al., we do not modify the architecture.

All models were trained using 4 NVIDIA H100 GPUs. VPT and VideoCAD required 4 days of training, while Pix2Act and Pearce et al. converged in 1 day. Quantitative results are summarized in Table 6.

We train VIDEOCADFORMER on the VIDEOCAD dataset with a 90/5/5 train/val/test split. At each step, the model autoregressively predicts the next action given a window of $k = 10$ past UI frames and actions. Outputs are treated as classification tasks and trained using cross-entropy loss with inverse-frequency class weighting. For pointer and typed parameters $(x, y, v)$, we apply soft targets over a $\pm 2$ bin window to account for geometric tolerance.

**Ablation Studies** To investigate the contribution of various modeling components and hyperparameters, we perform an extensive ablation study covering 16 model variants. These ablations isolate specific architectural and input conditioning factors that influence performance in long-horizon CAD action prediction. For these models, we train our ablations on a 90/5/5 split on 2500 samples of data.

First, we vary the model capacity, examining small versus large hidden dimensions (hidden size $\in \{512, 1024, 2048\}$) and feedforward projection sizes (feed forward $\in \{512, 1024, 2048\}$). Results show that increasing hidden size improves performance.

Table 6: Ablation results showing the impact of input types, image modalities, and architecture choices on command accuracy, parameter accuracy, and perfect action prediction across task difficulties.

| | $\mu_{\text{cmd}}$ (%) | $\mu_{\text{param}}$ (%) | Perfectly Predicted Actions (%) | | | | | |
| --- | --- | --- | --- | --- | --- | --- | --- | --- |
| | | | Mean | Max | Min | Short | Medium | Long |
| Base model | 94.90 | 71.55 | 79.27 | 97.21 | 54.24 | 83.88 | 76.94 | 76.03 |
| **Model Inputs** | | | | | | | | |
| No action | 88.92 | 65.11 | 73.11 | 96.59 | 40.34 | 81.15 | 68.07 | 68.52 |
| No state | 94.34 | 68.07 | 77.86 | 97.73 | 56.36 | 82.90 | 75.76 | 73.85 |
| No action and state | 17.35 | 3.80 | 5.59 | 25.52 | 0.00 | 2.30 | 7.78 | 7.32 |
| No timestep | 95.03 | 70.97 | 79.39 | 95.81 | 57.58 | 83.85 | 77.52 | 75.85 |
| Color jitter | 94.82 | 71.03 | 79.24 | 96.65 | 51.53 | 84.03 | 77.00 | 75.68 |
| **Images** | | | | | | | | |
| Base model (kernel-generated image) | 94.90 | 71.55 | 79.27 | 97.21 | 54.24 | 83.88 | 76.94 | 76.03 |
| Realistic Images | 94.05 | 68.50 | 77.62 | 95.45 | 56.57 | 83.20 | 74.86 | 73.64 |
| Sketches | 95.04 | 71.32 | 79.31 | 98.32 | 59.39 | 83.44 | 77.44 | 76.20 |
| **Window Size** | | | | | | | | |
| Past 5 actions and states | 94.97 | 72.22 | 79.86 | 97.73 | 58.64 | 84.49 | 77.78 | 76.35 |
| Base model (Past 10 actions and states) | 94.90 | 71.55 | 79.27 | 97.21 | 54.24 | 83.88 | 76.94 | 76.03 |
| Past 15 actions and states | 94.17 | 69.10 | 77.88 | 95.45 | 54.24 | 82.90 | 75.72 | 73.96 |
| **Feed Forward (FF.) Dim** | | | | | | | | |
| FF. dim=512 | 94.63 | 70.38 | 79.08 | 96.59 | 55.93 | 84.04 | 76.80 | 75.36 |
| Base model (FF. dim=1024) | 94.90 | 71.55 | 79.27 | 97.21 | 54.24 | 83.88 | 76.94 | 76.03 |
| FF. dim=2048 | 95.04 | 71.56 | 79.82 | 96.97 | 60.68 | 84.84 | 77.35 | 76.23 |
| **Hidden Size** | | | | | | | | |
| Hidden size=512 | 94.32 | 70.02 | 78.56 | 97.73 | 57.97 | 83.63 | 76.02 | 74.98 |
| Base model (Hidden size = 1024) | 94.90 | 71.55 | 79.27 | 97.21 | 54.24 | 83.88 | 76.94 | 76.03 |
| Hidden size=2048 | 95.11 | 71.49 | 79.67 | 96.79 | 58.38 | 83.85 | 77.74 | 76.55 |
| **Nhead** | | | | | | | | |
| Nhead=4 | 94.54 | 70.06 | 78.84 | 96.37 | 57.29 | 83.56 | 76.99 | 74.96 |
| Base model (Nhead = 8) | 94.90 | 71.55 | 79.27 | 97.21 | 54.24 | 83.88 | 76.94 | 76.03 |
| Nhead=16 | 95.70 | 72.19 | 79.96 | 98.04 | 62.37 | 84.43 | 77.51 | 77.03 |

Next, we assess the temporal window and attention configuration. Reducing or increasing the attention window (window size $\in \{5, 10, 15\}$) and varying transformer depth and heads (nhead $\in \{4, 8, 16\}$) allows us to probe the role of long-range context. Similar to model capacity, we observe increasing gains as we increase model capacity.

We also analyze the impact of input conditioning, disabling one or more context signals: past actions, past states, and timestep embeddings. Removing either past states or past actions significantly degrades performance, underscoring the importance of sequence memory in CAD modeling. The worst-performing variant disables both, indicating that VIDEOCADFORMER's success hinges on leveraging temporal continuity.

Additionally, we introduce a color jitter variant to test robustness to visual perturbations. Adding jitter slightly degrades performance, suggesting some sensitivity to pixel-space distortions and motivating future robustness strategies.

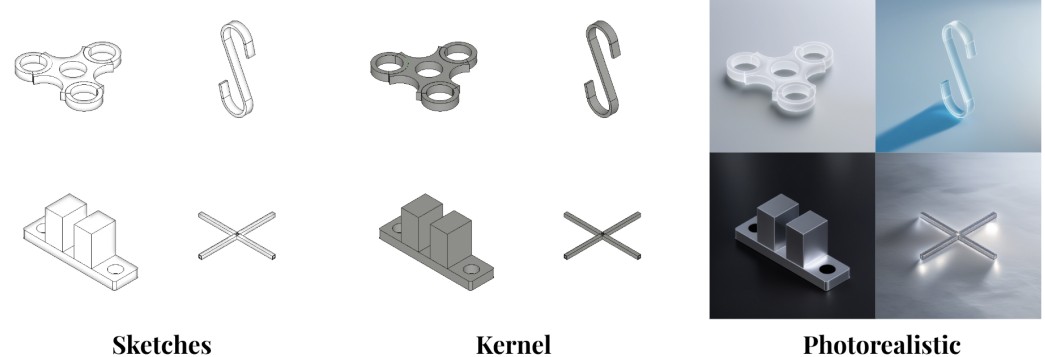

**Sketches**          **Kernel**          **Photorealistic**

Figure 8: Comparison of different CAD image inputs. From left to right: Sketches, Kernel and Realistic

Finally, we explore VIDEOCADFORMER's ability to generalize to different image types such as realistic images and sketches (Figure 8). We first generated images from a DeepCAD sequence using a geometry kernel. From this kernel-generated image, we constructed photo-realistic images by passing the kernel-generated CAD images into ControlNet [45] . The sketches are generated by applying canny line detection to the kernel-generated CAD images and then applying a Gaussian blur. Among the three, the model performs best on sketches, followed by kernel-generated images, and performs worst on photorealistic images, suggesting increased abstraction aids generalization.

We train a baseline model to act as a point of reference. The architectural parameters of the baseline model are shown in the table. Additionally, the baseline model has access to all inputs and are trained on grayscale isometric views of CAD images.

Overall, these ablations demonstrate that model performance is heavily influenced by temporal conditioning, architecture depth, and memory of past actions/states. These insights can guide the design of future agents operating in long-horizon, precision-driven software environments.

## C   Evaluation Metrics for CAD Generation

**Chamfer Distance after PCA Alignment.**   To quantitatively evaluate the geometric similarity between generated and ground truth CAD models, we begin by sampling point clouds uniformly from the surface meshes of each model. Let $P \subset \mathbb{R}^3$ denote the ground truth point cloud and $\hat{P} \subset \mathbb{R}^3$ denote the point cloud sampled from the generated CAD model. Because these models may differ in scale, orientation, or position, we first apply a similarity transformation to $\hat{P}$ that aligns it with $P$ before computing the metric.

This transformation consists of a rotation $R \in \mathrm{SO}(3)$, uniform scale $s \in \mathbb{R}$, and translation $t \in \mathbb{R}^3$, obtained via principal component analysis (PCA) alignment and RMS-based scale matching. Among all $48$ possible PCA axis permutations and sign flips, we select the transformation that minimizes the Chamfer Distance between the aligned prediction and the ground truth.

Formally, we define the aligned predicted point cloud as:

$$\hat{P}_{\text{aligned}} = \left\{ sR\hat{p} + t \,\middle|\, \hat{p} \in \hat{P} \right\},$$

and compute the symmetric Chamfer Distance as:

$$\mathrm{CD}(P, \hat{P}_{\text{aligned}}) = \frac{1}{|P|} \sum_{p \in P} \min_{\hat{p} \in \hat{P}_{\text{aligned}}} \|p - \hat{p}\|^2 + \frac{1}{|\hat{P}_{\text{aligned}}|} \sum_{\hat{p} \in \hat{P}_{\text{aligned}}} \min_{p \in P} \|\hat{p} - p\|^2.$$

We report the mean Chamfer Distance after this alignment as a measure of geometric fidelity.

**Command and Parameter Accuracy.**

$$\mu_{\text{cmd}} = \frac{1}{T} \sum_{t=1}^{T} \mathbb{1}[\hat{c}_t = c_t], \quad \mu_{\text{param}} = \frac{1}{T} \sum_{t=1}^{T} \frac{1}{d_t} \sum_{i=1}^{d_t} \mathbb{1}[\hat{p}_t^{(i)} = p_t^{(i)}] \cdot \mathbb{1}[\hat{c}_t = c_t]$$

# D  Uncertainty Quantification and Structural Fidelity of VideoCADFormer

**Measured Uncertainty Validates Significance of Results.** To quantify uncertainty, we estimate the standard error of the success rate using:

$$\sigma = \sqrt{\frac{p(1-p)}{n}}, \quad n = 200$$

For the **Medium-length** category:

$$\textbf{VPT:} \quad p = 0.364 \Rightarrow \sigma \approx 3.4\%$$
$$\textbf{VideoCADFormer:} \quad p = 0.379 \Rightarrow \sigma \approx 3.4\%$$

**VideoCADFormer Excels in Structural Fidelity and Robustness.** The modest difference between VPT and VideoCADFormer in medium-length sequences falls within statistical uncertainty, even after increasing the sample size. However, our model consistently outperforms across short and long sequences, achieving lower Chamfer Distance and generating fewer invalid CAD models across all sequence lengths.

We hypothesize that VPT tends to produce sequences that initially resemble the ground truth, performing adequately on medium-complexity cases. As sequence complexity increases (*Long* category), however, VPT fails to preserve structure and parametric correctness, leading to sharp drops in geometric fidelity. In contrast, VideoCADFormer maintains spatial and parametric consistency over longer horizons, demonstrating stronger generalization and robustness.

# E Image-Conditioned Generated CAD models using VideoCADFormer

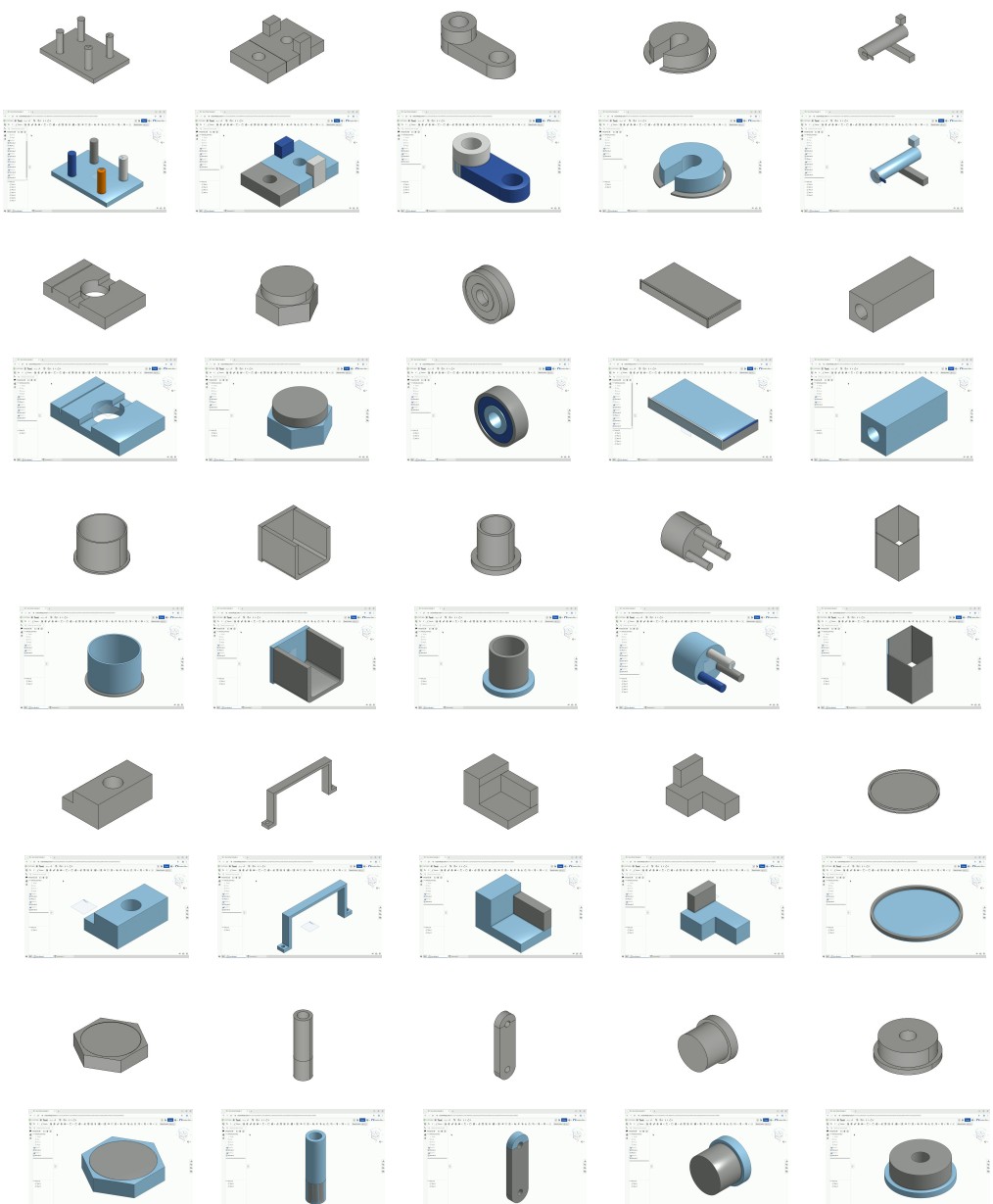

Figure 9: Some perfect samples generated using our trained model on Onshape

# F   Failure Analysis of VideoCADFormer

While our model demonstrates robust 3D reasoning from low-level UI actions, several failure cases persist. The most common error arises from inaccurate $(x, y)$ coordinate predictions during sketching, which often produce open loops that prevent successful extrusion. In some cases, these predictions result in closed loops with slight geometric inaccuracies—yielding extrusions that produce shapes visually similar to the target but with subtle deformations. These failures, as illustrated in Figure 10, are typically correctable with minor user intervention. Another frequent source of error is misclassification between line and arc primitives, especially in cases where curvature is visually ambiguous in the rendered images. These issues highlight the difficulty of resolving fine-grained geometric distinctions from image supervision alone and suggest that incorporating topological constraints or fine-tuning with interactive feedback could further improve model robustness.

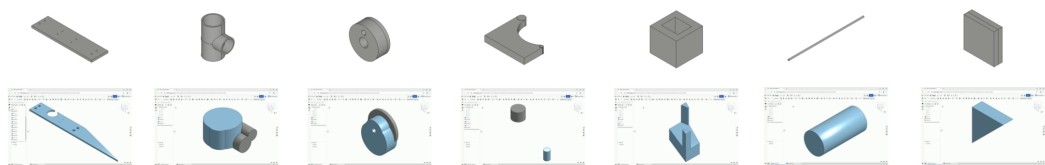

Figure 10: Failure cases

# G  Conducting a Failure Analysis on LLM Agents on CAD Design Tasks

Our current failure analysis of LLM agents is qualitative, as all of the tested agents were unable to make meaningful progress during evaluation. We have enclosed several *thought trace* samples to better illustrate the observed failure modes. As these models improve, we anticipate that more quantitative benchmarking will become meaningful.

We identify two common failure modes: (1) a lack of domain understanding of Onshape, and (2) grounding or perception issues.

### Addressing Domain Understanding Challenges

For issues pertaining to limited Onshape understanding, we commonly observe traces such as the following:

```
Action: It looks like my previous attempt to click the "Restore" button failed
because it might not have been correctly identified. Given that the current
context is focused on the "Sketch 1" with the message that it has been canceled,
the next logical step is to see if I can find the "Restore" button again or use
another strategy to re-engage with the sketch. I'll check for available buttons
or features related to the sketch in the current interface. Since there's no
immediate visible button to "Restore," I might try searching for features related
to the sketch or navigating through the interface. Next, I will scroll to see if
more options or buttons become visible.
scroll(0, 100)
```

In this case, the model attempted to restore a sketch by identifying alternative UI elements. However, it failed to recognize that the correct action was simply `Ctrl+Z`, instead issuing a scroll command on the sketch plane of the UI.

### Addressing Grounding Challenges

For issues related to grounding, we observe traces such as:

```
Action: Given the current context and the recurring issues with previous actions,
it's important to strategize the next step carefully:
1. Challenges Encountered: There have been errors when attempting to click
specific features, indicating either hidden elements or overlapping actions.
2. Current Focus: I'm in the Onshape part studio, where the workspace is largely
empty, with no parts created and the "Origin" feature hidden.
3. Goal: I need to find a way to effectively manipulate the sketch or geometry.

Strategy:
To move forward, it seems prudent to start by resetting my focus within the
workspace. Ensuring that my interaction is centered correctly might open up
further possibilities for manipulating features.

Next Action:
I will attempt to click on the "Sketch" section to see if that allows me to
access or reveal any relevant tools or options moving forward.
click("34")
```

Here, the model attempts to select the sketch section by referencing an internal element ID. However, in this case, `"34"` corresponds to a navigation bar element, not the intended sketch section, highlighting a fundamental grounding failure between perceived and actual interface elements.

# H Benchmarking LLMs' 3D Reasoning Capabilities

To assess LLMs' capabilities in spatial reasoning within CAD environments, we construct a synthetic Visual Question Answering (VQA) benchmark using the VIDEOCAD dataset. Each question is designed to probe specific aspects of geometric, temporal, and procedural reasoning essential for understanding 3D modeling workflows. The benchmark spans the following questions:

Table 7: Questions from our CAD-VQA dataset.

| Evaluation Type | Question Content | Choice Type |
|---|---|---|
| Extrusion Shape Prediction | You are given a completed sketch in Onshape. If the next command is Extrude, which image among the following will result from it? | Select correct image from 4 options |
| Num. Extrusion Estimation | How many extrusions were used in the provided CAD image? | Multiple choice (integers) |
| Extrusion Difference Prediction | You are given two extrusions for the same CAD model and the image of the CAD model. The second extrusion happens later than the first. Is the second extrusion deeper than the first? | Binary (Yes/No) |
| Sketch Ordering | Given these sketches from the video, order them to build the CAD object. | Identify correct sequence of sketches |
| Sketch Identification | Given an isometric view of a CAD model, select a sketch that was used to build this shape. | Select correct image from 4 options |
| Plane Identification | Given the following sketch and CAD image, which plane are you currently looking at? | Multiple choice: Top / Front / Right |
| CAD Primitive Identification | Which frame best matches the description of a given CAD primitive (arc, line, circle, extrude)? | Select correct image from 3 options |
| Sequence Prediction | What is the next primitive to draw (e.g., line, arc, etc.) given this CAD image and UI image? | Categorical multiple choice |
| Video Frame Sequencing | You are given 3 frames from the same video. What is the order of the frames? | Permutation over 3 items (6 options) |
| Hole Detection | Given this CAD image, is there a hole? | Binary (Yes/No) |
| Symmetry Detection | You are given an image of a CAD model. Across which planes is this CAD model symmetric? | Permutation over 3 planes (x,y,z) for a total of 8 options |

| Evaluation Type | gpt-4.1 [38] | claude-3-7 [39] | qwen2.5-vl [40] | o3 [41] | gemini-2.5 [42] | random |
|---|---|---|---|---|---|---|
| Extrusion Shape Prediction | 27.0 | 22.5 | 19.5 | 22.5 | 25.5 | **29.0** |
| Number of Extrusion Estimation | **47.0** | 37.5 | **47.0** | 45.0 | 38.0 | 21.4 |
| Extrusion Difference Prediction | 73.5 | **80.0** | 62.0 | 71.5 | 71.0 | 49.1 |
| Sketch Ordering | 37.5 | 29.5 | 41.0 | 37.0 | **61.0** | 35.0 |
| Sketch Identification | **62.0** | 48.5 | 43.5 | 48.5 | 59.0 | 21.5 |
| Plane Identification | 87.0 | 86.5 | 86.0 | **91.5** | 89.5 | 34.0 |
| CAD Primitive Identification | **84.1** | 80.1 | 78.6 | 75.1 | 82.6 | 40.3 |
| Sequence Prediction | **81.0** | 68.0 | 70.5 | 77.5 | 79.2 | 36.5 |
| Video Frame Sequencing | 36.0 | 23.0 | 32.5 | **80.0** | 73.2 | 17.4 |
| Hole Detection | **92.0** | 53.5 | 79.5 | 88.4 | 81.0 | 50.0 |
| Symmetry Detection | 18.5 | 19.0 | 12.0 | **27.9** | 27.0 | 12.5 |

Table 8: Performance of various vision-language models on the CAD-VQA benchmark across 11 evaluation tasks. Metrics are accuracy (%).

Collectively, these questions exercise geometric recognition, viewpoint awareness, forward simulation, temporal planning, and numeric inference — topics that are rarely interrogated simultaneously by existing VQA benchmarks. We generated 200 samples for each question for a total of 2,200 samples, though this question generation process can be easily scaled to the entire dataset. We provide the full benchmarking results on the questions below (Table 8).

# I   Positioning VideoCAD Within the Landscape of GUI Interaction Datasets

| Environment | # Samples | Time Horizon | 3D Reasoning | Precise Elements | Avg. # Elements |
|---|---|---|---|---|---|
| OSWorld | 369 | 15* | ✗ | ✓ | – |
| Mind2Web | 2,350 | 7.3 | ✗ | ✗ | 1,135 |
| WebArena | 812 | – | ✗ | ✗ | – |
| VisualWebArena | 910 | 35* | ✗ | ✗ | – |
| TheAgentCompany | 175 | 40 | ✗ | ✗ | – |
| WorkArena | 33 | 15 | ✗ | ✓ | – |
| WebShop | 12,000 | 11.3 | ✗ | ✗ | 38 |
| OmniAct | 9,802 | – | ✗ | ✓ | – |
| WebLinx | 2,337 | 43 | ✗ | ✗ | 1,849 |
| AITW | **715,142** | 6.5 | ✗ | ✗ | – |
| MMINA | 1,050 | 12.9 | ✗ | ✓ | 601 |
| MetaGUI | 1,125 | – | ✗ | ✗ | 79 |
| PixelHelp | 187 | 4.2 | ✗ | ✗ | – |
| AndroidWorld | 116 | 18.1 | ✗ | ✓ | – |
| AgentStudio | 304 | 30* | ✗ | ✓ | – |
| MoTIF | 4,707 | 4.4 | ✗ | ✗ | 188 |
| AndroidArena | 116 | 11.4 | ✗ | ✗ | – |
| WindowsAgentArena | 154 | 8.1 | ✗ | ✓ | – |
| MiniWoB++ | 125 | 3.6 | ✓ | ✗ | 28 |
| GUI-WORLD | 12,379 | 10.97 | ✓ | ✓ | – |
| **VideoCAD** | 41,005 | **186** | ✓ | ✓ | **6,740** |

Table 9: Full Comparison of multi-environment benchmarks for GUI interaction

* The max is used instead of the average as the average is not reported

The description of the metrics used to compare dataset complexity is explained in the following: **# Samples:** number of samples in the dataset, **Time Horizon:** number of UI interactions needed to complete the task, **Requires 3D Reasoning:** we look through every app used in each dataset. We determine whether the app requires the agent to reason about 3D coordinates to manipulate UI-state. **Contains Precise Element:** many benchmarks ground agents by leveraging the Document Object Model (DOM), enabling agents to select UI elements without needing to specify xy coordinates. However, some elements such as canvas elements cannot be interacted via the DOM. We define a precise element as an element that requires an agent to manipulate via xy coordinates, which require a higher level of reasoning as agents must rely on spatial and visual reasoning skills to click on the correct buttons. To check whether a benchmark requires a precise element, we look through every app used in each dataset and determine whether the agent is required to interact with a precise element such as a canvas, **Average UI elements:** To show that Onshape's complexity as a GUI, we measure the number of elements as described in Mind2Web. We only report the average number of elements for datasets that include the HTML tree in the dataset.

# J   Dataset Statistics

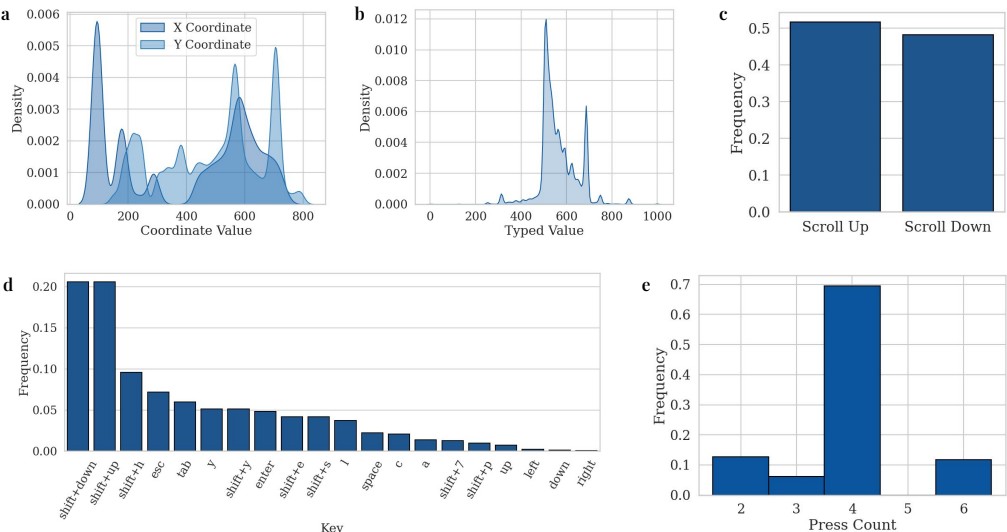

Figure 11: **(a)** Overlaid kernel density estimates for the X- and Y-coordinate distributions of mouse movements. **(b)** Kernel density of numeric values entered via the "Type" action. **(c)** Relative frequencies of scroll directions (up vs. down). **(d)** Frequency of individual key presses, sorted from most to least common. **(e)** Histogram of the number of times the "tab" key is pressed.

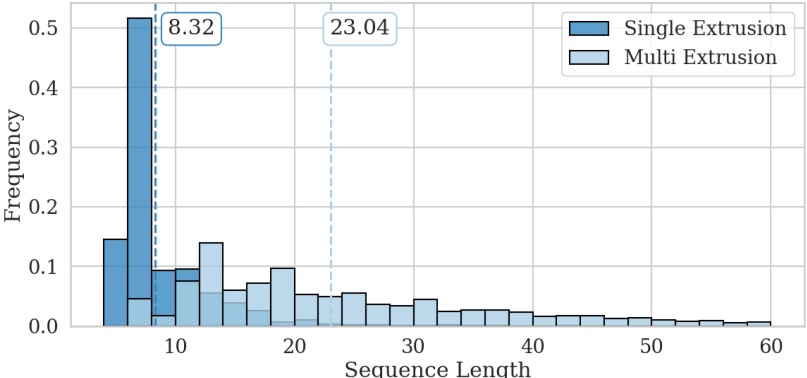

Figure 12: **Sequence length distribution of DeepCAD samples.** Histogram showing the normalized frequency of sequence lengths for single-extrusion (dark blue) and multi-extrusion (light blue) CAD sequences. The vertical dashed lines mark the mean sequence lengths (8.32 and 23.04, respectively).

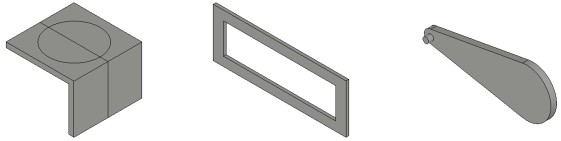

(a) 25th percentile samples; sequence length 109.

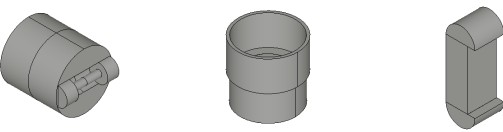

(b) 50th percentile samples; sequence length 157.

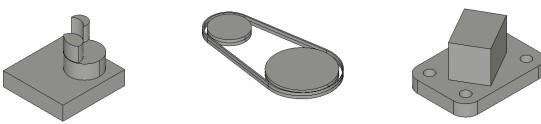

(c) 61st percentile samples; sequence length 186.

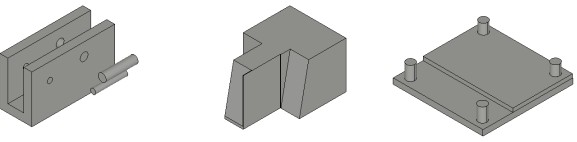

(d) 75th percentile samples; sequence length 239.

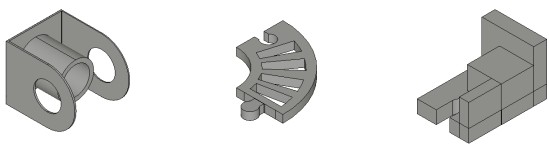

(e) 90th percentile samples; sequence length 337.

Figure 13: **Representative CAD models across sequence complexity percentiles.** For each percentile $p \in \{25, 50, 61, 75, 90\}$, we show three random samples. These examples illustrate the increasing structural complexity and diversity across the dataset distribution.

## K   Dataset Generation Procedure

CAD uses a sequence of parametric commands to create the shape step by step from scratch. Although there are a lot of CAD operations in existing commercial platforms, the two most widely used CAD operations are sketch and extrusion [8]. The purpose of the sketch command is to create the 2D sketch on a plane and then the extrusion command extrudes the sketch to create a 3D shape. To create complex and realistic 3D shapes, multiple sketches on different planes and multiple extrusion operations are required.

In the original DeepCAD dataset, $39.87\%$ of the samples are multi-extrude samples which is $71,424$ out of 179,133 shapes. After a filtering process described in section 3.1, we are left with $41,005$ samples. For the CAD sketch dataset, we create a synthetic sketch image of each CAD model sketch dataset by taking the Canny edge of an isometric CAD image and then adding a Gaussian blur.

**Rule-Based UI Automated Method.**   We use a hybrid rule-based automated method to execute the instruction sequences within Onshape. The method uses: **Selenium** for DOM-level browser automation, such as opening sketch menus, toggling extrusion modes, or navigating dialog boxes. **PyAutoGUI** for low-level pixel-based interactions within the sketch canvas, including drawing curves and typing parameter values. Since Onshape does not expose a public sketching API, this low-level simulation is essential.

All (x, y) interactions are normalized to the $(0, 1)$ screen space to ensure scale invariance. A timeout-based retry mechanism (3 retries, 5s each) is used for all UI actions. Episodes are terminated after three consecutive failures—most frequently caused by interface lags or rendering issues. During execution, we record high-resolution video at 60 FPS and log every UI action with sub-second alignment, deploying the system across 64 cloud VMs. In total, our automated method successfully reconstructed 44,292 CAD models out of 71,424 attempts (62.0%).

**Screen Recording and Logging.**   While the automated CAD construction is executed, we record high-resolution screen captures at 60 FPS using a custom Python video recorder. Simultaneously, we log every executed UI action (mouse event, keyboard press, selection change), each tagged with its precise frame index. This produces action–video alignment at sub-second granularity. The system is deployed across 64 Google Cloud VMs using 'Xvfb' to simulate GUI displays. Videos and logs are streamed to Google Cloud Storage. In total, over 118 days of screen recordings were generated in under one week.

**Introducing Human-like Heuristics to VideoCAD**   To balance realism and learnability in the presence of long-horizon 3D reasoning tasks (see Section 3.3), we introduce a few lightweight human-inspired heuristics into VideoCAD. First, we add randomized delays (0.2–0.5 seconds) between actions—mimicking human hesitation. Additionally, surface selections during sketching are made by randomly sampling a point on the surface, rather than always selecting the center. Finally, to emulate the need for precision, the automated method performs zoom actions when small features are hard to select, replicating how humans zoom in to refine their input.

# L  CAD Construction from DeepCAD Sequences

This appendix describes the complete process used to convert DeepCAD sequences [8] into executable CAD modeling instructions for Onshape, forming the backbone of our VideoCAD dataset.

## L.1  General Process Overview

For each CAD model, we iterate through the DeepCAD sequence and construct a list of `extrusions`. Each extrusion is defined by a 2D `sketch`, which contains a list of `loops`, and each loop is a sequence of geometric primitives—`lines`, `arcs`, or `circles`. After constructing the sketch, we extract extrusion parameters and store the full operation as a structured command.

## L.2  DeepCAD Representation

The DeepCAD sequence [8] represents modeling operations as a sequence of the form:

$$[t_i, x, y, \alpha, f, r, \theta, \phi, \gamma, p_x, p_y, p_z, s, e_1, e_2, u, b]$$

Each segment corresponds to a command: - $t_i$: command type (0: line, 1: arc, 2: circle, 4: loop separator, 5: extrusion) - `x`, `y`: points - $\alpha, f$: arc angle and direction - `r`: radius (for circle) - $\theta, \phi, \gamma$: plane angles - $p_x, p_y, p_z$: sketch origin offset - `s`: sketch scale - `e`$_1$, `e`$_2$: extrusion extents - `u`, `b`: extrusion operation type and symmetry

## L.3  Normalization

All values from the DeepCAD sequences are in $[0, 255]$ and are normalized using:

$$\mathcal{N}(p) = \frac{p - 128}{128}$$

Used for coordinates, angles, plane offsets, and extrusion parameters.

## L.4  Plane Basis and Extrusion Plane Parameters

The sketch plane is defined by normalized angles $(\theta, \phi, \gamma)$ and a 3D offset $(p_x, p_y, p_z)$. Using:

$$(\theta, \phi, \gamma) = \pi \cdot \mathcal{N}([\theta, \phi, \gamma]), \quad \mathbf{o} = \mathcal{N}([p_x, p_y, p_z])$$

we compute an orthonormal basis $(\mathbf{n}, \mathbf{x}, \mathbf{y})$, where:

$$\mathbf{n} = \text{polar-to-cartesian}(\theta, \phi, \gamma), \quad \mathbf{y} = \mathbf{n} \times \mathbf{x}$$

The sketch plane is mapped to an integer ID:

$$\texttt{plane\_id} \in \{0 : \texttt{Right}, 1 : \texttt{Front}, 2 : \texttt{Top}\}$$

The final extrusion plane offset is:

$$\texttt{offset} = 0.5 \cdot \mathbf{o}[\texttt{plane\_id}]$$

## L.5  Point Transformation to Pixel Space

Every geometric point in a sketch (e.g., endpoints, centers, midpoints) is transformed into 2D pixel coordinates through the following steps:

1. **Normalization:**
$$\mathbf{p}_{\text{norm}} = \mathcal{N}([x, y]) = \frac{[x, y] - 128}{128}$$

2. **Rotation and Offset:**
$$\mathbf{p}_{\text{rot}} = R \cdot \mathbf{p}_{\text{norm}} \cdot s + \mathbf{o}$$
where $R = [\mathbf{x}, \mathbf{y}]^T$ is the rotation matrix derived from the sketch plane basis, $s$ is the global scale, and $\mathbf{o}$ is the origin offset.

3. **Projection:** We project the rotated 3D point to 2D by dropping the axis corresponding to the sketch plane's normal direction.

Define a binary vector $\texttt{mask} \in \{0,1\}^3$ that keeps the two in-plane components:

$$\texttt{mask}[i] = \begin{cases} 0 & \text{if } i = \texttt{plane\_id} \\ 1 & \text{otherwise} \end{cases}$$

The 2D projected point becomes:

$$\mathbf{p}_{\text{proj}} = \mathbf{p}_{\text{rot}}[\texttt{mask}]$$

4. **Pixel Alignment:**

$$\mathbf{p}_{\text{pixel}} = 0.5 \cdot \mathbf{p}_{\text{proj}} + \mathbf{C}$$

where $\mathbf{C}$ is the pixel-space origin of the Onshape canvas.

This transformation is consistently applied to all sketch entities including:

- Line endpoints
- Circle centers
- Arc midpoints, center, start and end points

## L.6 Primitive-Specific Parameter Computation

Each loop in the sketch is composed of geometric primitives: **lines**, **circles**, and **arcs**, each defined by a set of transformed parameters. Below, we define each primitive mathematically using pixel-space coordinates.

### Lines

A line segment is defined by its start and end points. Let $\mathbf{p}_{\text{start}}, \mathbf{p}_{\text{end}} \in \mathbb{R}^2$ be the 2D pixel coordinates after projection:

$$\texttt{Line} = \{\mathbf{p}_{\text{start}}, \mathbf{p}_{\text{end}}\}$$

Each point is obtained by:

$$\mathbf{p} = 0.5 \cdot (R \cdot (\mathcal{N}([x,y]) \cdot s) + \mathbf{o})[\texttt{mask}] + \mathbf{C}$$

### Circles

A circle is defined by its center and radius. Let $\mathbf{c} \in \mathbb{R}^2$ be the projected center and $r$ the scaled radius:

$$r = \frac{r_{\text{seq}}}{128} \cdot s \cdot 0.5$$

$$\texttt{Circle} = \{\mathbf{c}, r\}, \quad \text{with } \mathbf{c} = 0.5 \cdot (R \cdot (\mathcal{N}([x,y]) \cdot s) + \mathbf{o})[\texttt{mask}] + \mathbf{C}$$

### Arcs

An arc is defined by its start point $\mathbf{p}_{\text{start}}$, end point $\mathbf{p}_{\text{end}}$, center $\mathbf{c}_{\text{arc}}$, midpoint $\mathbf{p}_{\text{mid}}$, and radius $r$.

**Angle and Direction.**

$$\alpha = 180 \cdot \frac{\alpha_{\text{seq}}}{128}, \quad f \in \{0,1\}$$

**Chord Geometry.**

$$\mathbf{v} = \mathbf{p}_{\text{end}} - \mathbf{p}_{\text{start}}, \quad L = \|\mathbf{v}\|, \quad \mathbf{p}_{\text{chord}} = \frac{\mathbf{p}_{\text{start}} + \mathbf{p}_{\text{end}}}{2}$$

**Radius and Offset.**

$$r = \frac{L}{2\sin(\alpha/2 \cdot \pi/180)}, \quad h = \sqrt{r^2 - (L/2)^2}$$

**Center and Midpoint.**

$$\mathbf{v}_\perp = \frac{[-v_y, v_x]}{\|\mathbf{v}\|}, \quad \mathbf{c}_{\text{arc}} = \begin{cases} \mathbf{p}_{\text{chord}} + h \cdot \mathbf{v}_\perp, & \text{if } f = 1 \\ \mathbf{p}_{\text{chord}} - h \cdot \mathbf{v}_\perp, & \text{if } f = 0 \end{cases}$$

The angular midpoint is computed as:

$$\theta_{\text{start}} = \text{atan2}(p_{\text{start},y} - c_y, p_{\text{start},x} - c_x)$$

$$\theta_{\text{mid}} = \theta_{\text{start}} \pm \frac{\alpha}{2} \cdot \frac{\pi}{180}$$

$$\mathbf{p}_{\text{mid}} = \mathbf{c}_{\text{arc}} + r \cdot \begin{bmatrix} \cos(\theta_{\text{mid}}) \\ \sin(\theta_{\text{mid}}) \end{bmatrix}$$

**Representation.**

$$\text{Arc} = \{\mathbf{p}_{\text{start}}, \ \mathbf{p}_{\text{end}}, \ \mathbf{c}_{\text{arc}}, \ \mathbf{p}_{\text{mid}}, \ r\}$$

All points are transformed to pixel coordinates using the transformation pipeline from Section L.

## L.7 Extrusion Parameters

Extrusion parameters are computed as:

$$u \in \{0 : \text{new}, 1 : \text{remove}, 2 : \text{union}\}, \quad b \in \{0 : \text{one-sided}, 1 : \text{symmetric}, 2 : \text{two-sided}\}$$

$$e_1 = 0.5 \cdot \mathcal{N}(e_1), \quad e_2 = 0.5 \cdot \mathcal{N}(e_2)$$

## L.8 Final Representation

Each extrusion is encoded as:

$$\text{Extrusion} = \{\texttt{plane\_id}, \ \texttt{offset}, \ u, \ e_1, \ e_2, \ b, \ \texttt{profile}\}$$

Where `profile` is a list of `loops`, and each loop is a list of geometric primitives: lines, arcs, and circles with parameters as computed above.

# M    Detailed Onshape UI Action Procedure

This appendix provides a thorough description of our automated CAD modeling procedure within Onshape, driven by our hybrid UI interaction framework based on rule-based commands. The procedure systematically translates structured CAD command sequences (defined in Appendix L) into executable UI interactions within Onshape.

## M.1    Plane Creation Procedure

For sketches requiring custom-defined planes (offset from default planes), we follow these UI steps:

1. Activate the plane creation tool by clicking the plane icon.
2. Select one of the default reference planes (`Top, Front, Right`).
3. Navigate to the offset text box via successive presses of the `Tab` key.
4. Enter the desired offset value numerically.
5. Click the offset direction arrow to define plane offset orientation.
6. Finalize plane creation by pressing `Enter`.

## M.2    Sketch Creation and Loop Building Procedure

Each sketch comprises loops defined by geometric primitives: lines, arcs, and circles. These loops are drawn via the following UI actions:

**General Sketch Setup:**

1. Start a new sketch by pressing `Shift+S`.
2. Select the appropriate sketch plane (custom or default).

**Drawing Geometric Primitives:**

- **Line**: Press L, then click to specify start and end points.
- **Circle**: Press C, click to set the center point, move cursor outward to specify radius, then click to finalize.
- **Arc (3-point)**: Press A, sequentially click to set start, end, and midpoint.

**Constraint Management:**    To ensure geometric accuracy, constraints are toggled dynamically:

- Press `Shift` key down to deactivate constraints.
- Press `Shift` key up to activate constraints (primarily when connecting loop endpoints).

**Command Completion:**

- Exit current geometric command (line, arc, circle) using `Esc`.

## M.3    Visibility and Navigation

Visibility and navigation are managed to maintain clarity and prevent interference from previously drawn elements:

- Toggle visibility of planes: `Shift+P`.
- Toggle visibility of sketches: `Shift+H`.
- Hide/show previously built parts: `Y/Shift+Y`.
- Navigate between default planes (`Top, Front, Right`) using key sequences: `Shift (down) → + → (Up/Right/Down/Left) → Shift (up)`.

## M.4 Extrusion Execution Procedure

Extrusions convert sketch loops into 3D features through these UI steps:

1. Select sketch region(s) intended for extrusion.
2. Activate extrusion using `Shift+E`.
3. Choose extrusion type (`New` for adding material, `Remove` for material removal).
4. Navigate (using `Tab`) to set the extrusion depth numerically.
5. Navigate further (using `Tab`) to the symmetric option checkbox; press `Space` to toggle symmetry on/off.
6. If performing material removal (`Remove`), navigate to and activate the "Merge with all" checkbox.
7. Finalize extrusion by pressing `Enter`.

## M.5 End-of-Sequence Token

We use the hotkey combination `Shift+7` to set the camera view to an isometric perspective. This action signifies the completion of the CAD modeling sequence and serves as our end-of-sequence token.

# N    Ablation on Vision Models for CAD Image Similarity

Table 10: Comparison of CLIP and DINOv2 for CAD image similarity over 400 samples.

| Metric | CLIP | DINOv2 |
|---|---|---|
| Mean of correct scores | 0.8576 | 0.7923 |
| Median of correct scores | 0.8587 | 0.8022 |
| Std Dev of correct scores | 0.0250 | 0.0488 |
| Mean of incorrect scores | 0.7931 | 0.6002 |
| Median of incorrect scores | 0.7947 | 0.6014 |
| Std Dev of incorrect scores | 0.0167 | 0.0440 |
| Average rank of correct match | 18.24 | **2.21** |
| Median rank of correct match | 6.00 | **1.00** |
| Std Dev of rank | 31.69 | **3.08** |

We conduct an ablation comparing **DINOv2** and **CLIP** as image encoders for CAD shape comparison. For each of 400 query samples, we compute the cosine similarity between features extracted from a rendered UI isometric image and a set of 400 ground-truth isometric CAD images (including the correct one). We define the *correct score* as the similarity between the UI image and its associated CAD image, and the *incorrect scores* as the similarities with all other CAD images in the set. The rank of the correct match is computed by ranking all 400 similarity scores in descending order.

Table 10 reports the results. Although CLIP yields slightly higher correct match scores on average, its incorrect scores are also high, indicating limited discriminative power for geometric retrieval. In contrast, DINOv2 achieves significantly lower similarity scores for incorrect matches (mean of 0.6002 vs. 0.7931 for CLIP), resulting in more robust separation and better retrieval accuracy. This is reflected in the rank-based metrics: DINOv2 achieves an average correct match rank of **2.21** and a median of **1.00**, far outperforming CLIP (18.24 and 6.00, respectively), with far less variance (std dev of 3.08 vs. 31.69).

This improvement can be attributed to the architectural and training differences between the models: **CLIP** is optimized for semantic alignment between text and image—learning "what is this"—whereas **DINOv2** is trained in a self-supervised manner to model fine-grained visual structure, capturing both local and global geometry. This makes DINOv2 naturally better suited for comparing CAD-like images, where shape and spatial configuration are more critical than semantic labels.

**Threshold Selection.** We also use these similarity scores to define a threshold for dataset quality control. Specifically, we set the threshold at 0.7, which corresponds to the average of the median correct score (0.8022) and median incorrect score (0.6014). This ensures a conservative yet robust filter for identifying high-quality image-CAD pairs.

## O Temporal Abstraction and Causal Structure in VideoCAD

**CAD Sequence Length vs. UI Sequence Length (Time Horizon).** The difference between the statistics in Figure 12 and Figure 3 arises from the distinct abstraction levels of the reported sequences. Figure 12 shows the distribution of *CAD sequence lengths* derived from the DEEPCAD dataset, representing the number of high-level design operations (e.g., *Line*, *Extrude*). These symbolic commands reflect semantic modeling intent and typically average around 23 operations per design.

In contrast, Figure 3 reports the *UI time horizon*, corresponding to the number of low-level user interactions (e.g., mouse clicks, key presses, and pointer movements) required to complete a CAD model in the interface. These interactions capture fine-grained execution detail and are substantially more numerous, particularly for complex or precise geometry. A single high-level CAD command, such as an extrusion, can involve dozens of low-level UI actions including plane selection, primitive drawing, parameter entry, and menu navigation. Consequently, CAD models with 20-30 symbolic steps often result in approximately 150-200 UI actions on average, explaining the difference in reported scales.

**Intermediate Modeling Processes.** VIDEOCAD explicitly captures intermediate modeling stages. Each trajectory is recorded as a multi-frame video synchronized with both low-level UI events and high-level CAD operations, providing aligned visual and symbolic representations of intermediate construction states. This structure enables models to learn not only final outcomes but also the step-by-step reasoning underlying geometric construction. A more complex example of intermediate states leading to a target CAD model is shown in Figure 14.

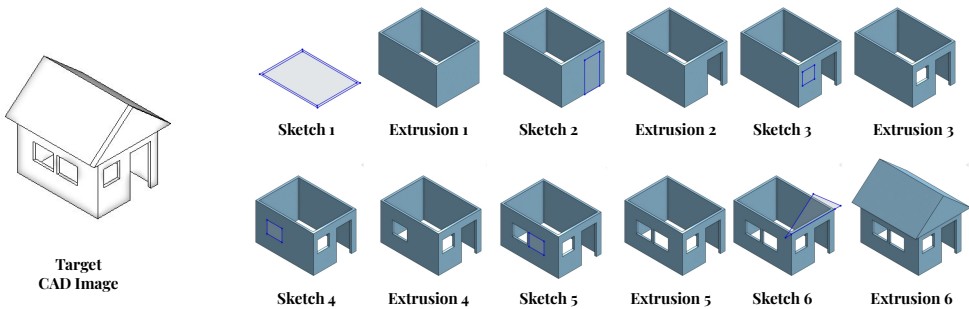

Figure 14: **Example of intermediate modeling stages in VIDEOCAD.** A sequence of snapshots illustrating the progressive construction of a CAD model through successive sketching and extrusion operations.

**Causal Dependencies.** CAD modeling exhibits strong causal dependencies, for example, a sketch must be defined before an extrusion can be performed. Our autoregressive transformer explicitly models these dependencies using causal masking and windowed attention (Section **??**), ensuring that each predicted action is conditioned on the full prior sequence and follows valid operation orderings.

**Permutable Actions.** Some modeling actions, such as drawing multiple disconnected primitives, can be executed in different orders without altering the final geometry. The current dataset provides a single reference trajectory per CAD model, assuming a unique canonical sequence. Extending this framework to handle permutation-invariant or multi-trajectory supervision is a promising future direction.

