# OpenReview forum: "VideoCAD: A Dataset and Model for Learning Long‑Horizon 3D CAD UI Interactions from Video"
_NeurIPS.cc/2025/Datasets_and_Benchmarks_Track — NeurIPS 2025 Datasets and Benchmarks Track poster_

### Official Review · Reviewer_1MxM · 2025-06-09

**Rating:** 5
**Confidence:** 3

**Summary:**

This paper presents VideoCAD, a novel synthetic dataset of 41K annotated CAD operation videos with UI interactions, addressing a critical gap in long-term, complex, precise UI agentic action learning. The work is technically sound, offers significant contributions (dataset, SOTA model, VQA benchmark), and demonstrates rigorous experimentation. While limitations exist, the dataset's potential for advancing UI agents, industrial product generation, and 3D reasoning justifies its significance.

**Additional Feedback:**

There is no additional feedback.

**Dataset Code Accessibility:**

Yes

**Dataset Code Comments:**

Data is available at https://dataverse.harvard.edu/dataset.xhtml?persistentId=doi:10.7910/DVN/WX8PCK&faces-redirect=true.
Code is available at https://github.com/BrandonMan123/VideoCAD.
However, the README is not complete at Github. It is suggested to complete it with detailed instrucion on how to access the dataset and reproduce your experiments.

**Ethical Comments:**

It is synthetic data, so there are no privacy risks and ethical considerations that need to be raised.

**Ethical Considerations:**

No, there are no or only very minor ethics concerns

**Final Justification:**

After reviewing the authors' rebuttal, I will keep my positive rating of 5, suggesting for acceptance.

**Limitations Weaknesses:**

1. Rule-based bot trajectories lack human variability (errors, corrections). While heuristics (random delays, zooming) help, real user data is still irreplaceable for developing robust models.
2. The data only include sketch-extrude workflows. It is a recognized limitation, but the dataset presents a valuable foundational step towards this research area.
3. LLM agent failure analysis is qualitative; quantifying error modes would add depth.
4. The proposed method discretizes parameters into 1000 classes. The potential impact of this discretization on performance and scalability warrants discussion, particularly concerning how the method would perform if scaled to significantly higher-dimensional or finer-grained parameter spaces. Will it still work?
5. For the evaluation metrics of offline close-loop execution performance, why do you design it in such a way, rather than using existing sequence edit distance or other sequence accuracy metrics?

**Strengths Contributions:**

1. It has the potential to be a high-impact dataset, as it addresses a critical research gap in 3D reasoning and precise UI agent development.
2. The VideoCADFormer and VideoCADQA benchmarks are also valuable to motivate future research.
3. The experiment and evaluation process is rigorous, providing valuable insights for other researchers.

---

> ### Author Rebuttal · Authors · 2025-07-30
>
> We thank the reviewer for the questions. We address the posed questions in order.
> ### **1. Incorporating Human Variability into VideoCAD**
> We agree that incorporating human variability—such as corrections, hesitations, and strategy shifts—is crucial for developing robust and generalizable UI agents. As acknowledged in Section 6, our roadmap explicitly includes incorporating human demonstration data, such as from YouTube tutorials, and collecting multiple trajectories per target design to reflect diverse strategies and behaviors (Section 6, points 1 and 4). Thus, while we agree that human data is ultimately irreplaceable for developing fully robust and generalizable UI agents, we view VideoCAD as a critical first step. It provides large-scale, fully observable, and precisely labeled sequences that enable foundational research and benchmarking. Our framework is designed to scale to human data, and we are actively pursuing these extensions as part of our future work.
>
>
> ### **2. Addressing Future Extensions of VideoCAD**
> As mentioned in a response to Reviewer 3’s first point, our goal in this first version is to establish a scalable, high-fidelity benchmark for long-horizon CAD interaction learning, grounded in a well-defined and interpretable action space. We plan to extend the dataset to more complex workflows in the future.
>
>
> ### **3. Conducting a Failure Analysis on LLM Agents on CAD Design Tasks**
> Our current failure analysis of LLM agents is qualitative as all of the LLM agents we tested were unable to make any meaningful progress during the evaluation. We have enclosed a few thought trace samples for your reference and we plan to include more in our final version of the paper. We anticipate that as these models get better, it would make more sense for more quantitative benchmarking.
> We find that the two common failure modes are a lack of domain understanding of Onshape and grounding/perception issues.
> ### Addressing domain understanding challenges
> For issues pertaining to a lack of Onshape understanding, we commonly observe traces such as the one below:
> ```
> Action: It looks like my previous attempt to click the "Restore" button failed because it might not have been correctly identified.
> Given that the current context is focused on the "Sketch 1" with the message that it has been canceled, the next logical step is to see if I can find the "Restore" button again or use another strategy to re-engage with the sketch.
> I'll check for available buttons or features related to the sketch in the current interface. Since there's no immediate visible button to "Restore," I might try searching for features related to the sketch or navigating through the interface.
> Next, I will scroll to see if more options or buttons become visible.
> ```scroll(0, 100)```
> ```
> In this case, the model was trying to restore a sketch, and it was trying to identify alternatives to restoring a sketch. The model failed to recognize the correct button to perform a restoration (which is just ctrl+z) and instead scrolled on the sketch plane of the UI.
> ### Addressing grounding challenges
> For issues related to grounding, we commonly observe traces such as the one below:
> ```
> Action: Given the current context and the recurring issues with previous actions, it’s important to strategize the next step carefully:
> 1. **Challenges Encountered**: There have been errors when attempting to click specific features, indicating either hidden elements or the potential overlap of actions.
> 2. **Current Focus**: I'm in the Onshape part studio, where the workspace is largely empty, with no parts created and the "Origin" feature hidden.
> 3. **Goal**: I need to find a way to effectively manipulate the sketch or geometry.
> ### Strategy:
> To move forward, it seems prudent to start by resetting my focus within the workspace. Ensuring that my interaction is centered correctly might open up further possibilities for manipulating features.
> ### Next Action:
> I will attempt to click on the "Sketch" section to see if that allows me to access or reveal any relevant tools or options moving forward.
> ```click("34")```
> ```
> The model is trying to click on the sketch section, and it does so by selecting the what it thinks is the correct element ID. However, “34” in this case corresponds to the navigation bar, which is completely different from the sketch section.
> ### **4. Addressing scalability issues for VideoCADFormer**
> We appreciate the reviewer’s concern regarding discretization and its impact on scalability. Our choice of 1000 discrete classes was a deliberate design decision informed by the precision requirements of CAD UI interaction and computational efficiency
> ### Screen Coordinate Precision:
>
>
> We operate the UI using normalized coordinates with three digits of precision (i.e., values ranging from 0.000 to 0.999). This makes 1000 discrete bins a natural and sufficient choice for representing pointer positions and other continuous parameters. We verified that this level of resolution enables accurate reconstruction of CAD models from ground-truth actions.
> ### Handling Fine-Grained Details via Zooming:
>
>
> Although the base resolution is fixed at 1000 classes, the environment supports zooming actions, allowing the agent to focus on specific regions and operate with increased effective resolution. This ensures that fine-grained details can be precisely specified without requiring a globally larger action space.
> ### Unified Learning Objective:
>
>
>  Discretizing all parameters allows us to cast the entire action prediction task—including spatial and scalar parameters—as a pure classification problem. This unifies the loss function, simplifies training, and improves model stability.
>
> ### **5. Explaining Design Decisions behind our evaluation metric**
>
> We thank the reviewer for the suggestion to consider existing sequence metrics such as edit distance. While metrics like Levenshtein, Hamming, and LCS are informative in domains such as text generation, they fail to capture the strict sequential dependencies and execution semantics required in CAD environments.
>
> ### Failure mode with edit-based distances (Levenshtein distance)
>
> Edit distance may assign the same score to qualitatively different sequences. Consider the ground truth sequence [move(a, b), move(c, d), move(a, b), click()]. Both [move(c, d), move(a, b), move(a, b), click()] and [move(a, b), move(c, d), click(), move(a, b)] yield a Levenshtein distance of 2, yet the latter significantly disrupts execution order. Similarly, Hamming distance fails to capture severity of functional deviation—for example, replacing a click with a zoom in [move, click, move, click] gives a low Hamming distance but causes critical task failure.
>
> ### Failure mode with subsequence-based distances
>
> Longest Common Subsequence (LCS) and Jaro–Winkler distance do not penalize early errors appropriately. In a sequence such as [move, click, move, click, move, click], a candidate like [click, click, move, click, move, click] can have a high LCS, yet early errors cascade and make the final CAD model invalid.
>
> These examples highlight that such sequence metrics do not reflect the execution fidelity necessary for deployment, where one incorrect step invalidates the downstream modeling process. Our chosen metric directly measures stability under full autoregressive rollout and is complemented by geometric Chamfer distance and invalid mesh rate to evaluate final output quality.

---

> > ### Comment · Reviewer_1MxM · 2025-08-05
> > **Maintain my positive rating.**
> >
> > I appreciate the revision. It addressed all of my concerns. I also appreciate the thought trace samples that the authors provided, which helped me with understanding the performance of current LLM agents. I am happy to maintain my positive rating.

---

### Official Review · Reviewer_YkWZ · 2025-06-16

**Rating:** 4
**Confidence:** 5

**Summary:**

The paper introduce VideoCAD, the first large-scale video dataset specifically designed to support learning and automation in a professional Computer-Aided Design (CAD) environment. Existing datasets have all limited use cases targeting only simple, short duration tasks in limited online or mobile applications, while this dataset is designed specifically for the complex, high-precision user interactions with long-horizon spatial reasoning found in CAD software. VideoCAD consists of more than 41k synthetic videos of CAD Modeling, created from human-authored designs, modeled by a rule-based bot in Onshape, a browser-based CAD tool. Each video is richly annotated with annotations of low-level UI interactions as well as high-level modeling operations, alongside detailed temporal and visual context.

Additionally, the paper introduce a transformer-based model called VideoCADFormer that predicted CAD UI actions, outperforming state-of-the-art behavior cloning methods, in both spatial action mapping, and geometric reconstruction. They also developed VideoCADQA, a visual question answering (VQA) benchmark sourced from the same dataset to evaluate whether large language models (LLMs) can reason about 3D geometry and understand logic in video sequences. Experimentation showed current multimodal LLMs performed poorly at CAD-related reasoning tasks, such as extrusion depth comparisons or feature recognition from sketches, revealing large limitations of capability in existing models.

**Dataset Code Accessibility:**

Partly

**Dataset Code Comments:**

The authors provided both the dataset and code, but the documentation is limited and lacks sufficient detail for easy reproduction. Clearer instructions, usage examples, and metadata descriptions would be needed to make the submission fully accessible and reproducible.

**Ethical Considerations:**

No, there are no or only very minor ethics concerns

**Final Justification:**

I would like to thank the authors for their detailed responses during the rebuttal period, which have successfully addressed my concerns. I recommend that this paper be accepted.

**Limitations Weaknesses:**

1. While CAD interaction datasets are especially valuable because they can capture entities for operations like chamfer and fillet--whose required geometric context is often invisible in traditional design history dataset(like Deepcad[1])--the current version of VideoCAD focuses only on sketch and extrusion workflows and does not support these more advanced operations. Nevertheless, the dataset provides a solid foundation for future expansion.

2. While the dataset is built entirely on Onshape--posing general concerns about cross-platform applicability--another issue lies in potential violations of Onshape’s Terms of Use. Section 4.a.ix prohibits:

> "Using any robot, spider, scraper or other automated means to access the Service, or use any data mining, data gathering or extraction method."

This clause broadly disallows any form of automated access to the Onshape platform, regardless of intent. This raises concerns about the long-term sustainability, reproducibility, and ethical standing of the dataset. Any future research relying on this approach risks violating platform restrictions.

[1] Rundi Wu, Chang Xiao, and Changxi Zheng. Deepcad: A deep generative network for computer aided design models. In Proceedings of the IEEE/CVF International Conference on Computer Vision, pages 6772–6782, 2021.

**Strengths Contributions:**

1. The paper fills an important gap in current UI datasets, by providing the first large-scale, high-fidelity dataset specifically for professional CAD software with long-horizon, 3D spatial reasoning. Previous datasets focused on simpler web/mobile applications do not encompass these more complex UI interactions.
2. The introduced model, VideoCADFormer, provides a strong baseline for CAD UI actions prediction, and also outperforms past behavior cloning methods in command accuracy and geometrical reconstruction results.
3. VideoCAD contains nearly 41k video samples, all of which are labeled with low-level UI interactions (e.g., clicks, mouse movements) as well as high-level modeling actions (e.g., extrusions, sketches).

---

> ### Author Rebuttal · Authors · 2025-07-31
>
> ### **1. VideoCAD easily extends to more complex CAD commands**
>  We acknowledge that CAD operations are complex and operations such as chamfer, fillet, loft, and sweep is an important direction. We first set out to build the VideoCAD dataset when we found that LLM agents were unable to complete even simple sketch-extrude workflows, which is why we focused on these workflows. As mentioned in Section 6, we plan to expand the number of operations in future iterations of this dataset. VideoCAD was designed to be extensible. The action schema, UI bot infrastructure, and modeling framework can naturally accommodate additional operations in future iterations. Our goal in this first version is to establish a scalable, high-fidelity benchmark for long-horizon CAD interaction learning, grounded in a well-defined and interpretable action space. This initial release lays the groundwork for such expansion by establishing a robust end-to-end pipeline for mapping procedural geometry to UI actions and evaluating model performance through geometric fidelity.
> Responsible Use and Compliance with Onshape Terms
> ### **2. Transparency in VideoCAD dataset generation**
> We have been aware of the Onshape policy while developing the dataset and would like to clarify several important points.
> We have presented VideoCAD work and its generation pipeline to the Onshape team and have also been in close communication with them on the broader topic of CAD datasets for AI research. Specifically, we presented our work to Jon Hirschtick (Co-founder of Onshape) and Mark Cheli (Director, Onshape Product Operations & Strategy). Additionally, we hosted the Onshape team at our institution and visited their headquarters to discuss this and other related work. The team is supportive of our VideoCAD work and has never raised any compliance issues. Furthermore, we continue to be close collaborators with Onshape and plan to further collaborate in the future on datasets enabling AI research on the topic. If required, we would be happy to add a statement from the Onshape team acknowledging permissible use to our Harvard dataverse page.
>
>
> ### Usage of only public datasets under MIT License:
>
>
> All CAD models used in VideoCAD were derived from public and freely licensed data (DeepCAD [1]), and no user data or private IP was accessed at any point. The Onshape platform is used strictly as a visual frontend to simulate CAD modeling actions. DeepCAD has been used in several studies from ICML 2025 and NeurIPS including CAD-Editor [2] and Text2CAD [3] respectively.
>
>
> ### Scope of data on front-end screen recording:
> Our dataset was generated through rule-based interaction with the front-end UI, using tools like Selenium and PyAutoGUI—tools often employed for accessibility testing and user interface prototyping. No scraping or reverse engineering was conducted on Onshape's backend services or internal APIs.
>
>
> While we acknowledge that many platforms (including YouTube, Amazon and CAD software) generally discourage automated access, we have prioritized transparency in the development of VideoCAD, adhered to fair-use principles for academic research, and exclusively used data under the permissive MIT license.
>
>
> ### **3. VideoCAD data is cross-platform generalizable**
> Part of our data annotation process involves recording the mouse position and keyboard actions. It is straightforward to map the coordinates of sketch or extrude buttons from one software to another. As a result, we address this in Section 6 by outlining our plan to extend the dataset to other CAD platforms such as Fusion 360 and FreeCAD. We selected Onshape due to its browser-based accessibility, educational and industry adoption, and ease of deployment across institutions. FreeCAD was initially considered, but its limited adoption risks reducing real-world relevance and downstream impact. To support future expansion, our codebase is modular and designed from the outset to support integration with other platforms.
>
>
>
> [2] Yu Yuan, Shizhao Sun, Qi Liu, and Jiang Bian, “CAD‑Editor: A Locate‑then‑Infill Framework with Automated Training Data Synthesis for Text‑Based CAD Editing,” Proceedings of the 2025 International Conference on Machine Learning (ICML 2025), July 2025
> [3] Mohammad Sadil Khan, Sankalp Sinha, Talha Uddin Sheikh, Didier Stricker, Sk Aziz Ali, and Muhammad Zeshan Afzal, “Text2CAD: Generating Sequential CAD Designs from Beginner-to-Expert Level Text Prompts,” Advances in Neural Information Processing Systems, vol. 37, pp. 7552–7579, 2024.

---

### Official Review · Reviewer_vz4V · 2025-06-21

**Rating:** 5
**Confidence:** 3

**Summary:**

This paper proposes a large-scale video dataset for learning UI interactions and 3D reasoning from CAD software.The dataset consists of more than 41K annotated video recordings of CAD operations, generated by the authors using an automated framework for collecting high-fidelity UI operation data from man-made CAD designs. Compared to existing datasets, VIDEOCAD provides a higher complexity of UI interaction learning for real-world engineering tasks over a time span 20 times longer than other datasets. The paper demonstrates two important downstream applications of VIDEOCAD: learning UI interactions from specialized precision 3D CAD tools, and designing Visual Question and Answer (VQA) benchmarks for assessing spatial reasoning and video comprehension capabilities of multimodal Large Language Models (LLMs). In addition, the authors present VIDEOCADFORMER-a state-of-the-art model for learning CAD interactions directly from video that outperforms multiple behavioral cloning baselines.Both VIDEOCADFORMER and VQA benchmarks derived from VIDEOCAD reveal current video-based UI understanding critical challenges, including the need for accurate action bases, multimodal and spatial reasoning, and long-horizon dependencies

**Dataset Code Accessibility:**

Yes

**Dataset Code Comments:**

https://github.com/BrandonMan123/VideoCAD

**Ethical Considerations:**

No, there are no or only very minor ethics concerns

**Final Justification:**

The author's rebuttal basically solved my confusion

**Limitations Weaknesses:**

1. What is the relationship between the sequence length in Fig. 3 b and the time horizon in Table 1? Why is it that most of the sequence lengths shown in b are 8.32, while Table 1 shows 186 time horizons for each task?
2. Which examples need to be so long e.g. 186 actions per task?Can you give some examples, the examples in the paper are very visual and will make it difficult for the reader to understand the difference in complexity between the proposed dataset and the existing dataset
3. For long-horizon tasks, is it possible to show intermediate processes in the dataset as well as in the experimental results? As well as whether there is a causal relationship between actions, whether there are some steps whose back and forth order does not affect the final result, and how the method and optimization process is handled in this case?

**Strengths Contributions:**

1. Compared with previous work, the dataset is larger, with richer samples, contains rich 3D reasoning, fine-grained elements, and is more valuable for the study of long-horizon dependent UI interaction work.
2. the paper is clearly written, the dataset collection process is clearly described, and the experiments are relatively adequate.

---

> ### Author Rebuttal · Authors · 2025-07-30
>
> We thank the reviewer for the questions. We address the posed questions in order.
> ### **1. Clarifying the relationship between sequence length and time horizon**
> The apparent discrepancy arises because Figure 3b and Table 1 report statistics over different types of sequences at different levels of abstraction.
> Figure 3b reports the distribution of CAD sequence lengths derived from the DeepCAD dataset, which captures the number of high-level CAD operations (e.g., sketch, extrusion). These are symbolic design commands that reflect semantic modeling intent—typically compact and averaging around 8.3 operations per design.
>
>
> In contrast, Table 1 reports the UI time horizon, which corresponds to the number of low-level UI interactions (e.g., mouse clicks, key presses, pointer moves) needed to complete a CAD model using a GUI. These are detailed actions recorded during UI execution (defined in Table 2) and are substantially more numerous, especially for fine-grained geometry or precise input.
>
>
> A single high-level CAD operation (like an extrusion) can involve dozens of low-level UI actions, such as selecting the sketch plane, drawing multiple primitives, typing parameter values, and navigating menus. Therefore, CAD models with 8–10 high-level steps often result in hundreds of UI actions, thus explaining the average time horizon of 186 in Table 1.
> We will clarify this distinction in the final version of the paper to avoid confusion between symbolic CAD command length and UI-level interaction length.
>
>
> ### **2. Visualizing the relation between sequence length and CAD model complexity**
> We understand that these metrics are highly visual and additional illustrations would help readers better understand how sequence lengths map to CAD model complexity. We will take note of this in the next revision where we will provide 3 images of sample CAD at the 25th, 50th, 75th and 90th percentiles as well as sequences of length 186 (61st percentile). Due to the photo-sharing limitations, we invite the reviewer to reference the first row of section VII of fig. 1 of the paper. We show the time horizon and their corresponding percentile for each CAD object on the top row from left to right:
>
>
> | Position        |  1 |  2 |  3 |  4 |  5 |  6 |
> |-----------------|---------|---------|---------|---------|---------|---------|
> | Percentile (\%) | 91.82   | 96.51   | 98.42   | 87.86   | 65.17   | 90.53   |
> | Length          | 358     | 422     | 466     | 317     | 199     | 343     |
>
>
>
>
>
>
> ### **3. Showing Intermediate Processes and Causal Relationships**
> We clarify the following:
> ### Intermediate Processes:
> Yes, our dataset explicitly includes intermediate steps. Each CAD modeling trajectory in VideoCAD is captured as a full-length multiframe video aligned with low-level UI actions and high-level CAD operations. These sequences provide both visual and symbolic traces of intermediate construction stages, which are used directly in training and evaluation.
> ### Causal Relationships:
> CAD modeling inherently involves strong causal dependencies—for example, sketches must be completed before extrusions can be executed. Our model respects these dependencies via an autoregressive transformer architecture with causal masking and windowed attention (Section 4), ensuring that each predicted action is conditioned on the full sequence history.
> ### Permutable Steps:
> We acknowledge that in some cases, certain actions (e.g., drawing multiple disconnected primitives) could be executed in different orders without affecting the final geometry. However, our current dataset only provides a single reference trajectory per CAD model, and our training objective assumes a unique action sequence. Modeling such permutation-invariant action spaces is an exciting direction for future work, potentially involving reinforcement learning or trajectory clustering methods.
> ### Handling in Optimization:
>
>
> Our current optimization process follows behavior cloning with a strict supervised loss on the reference sequence. While this enables precise imitation, it does not capture alternative valid execution paths. We note this limitation in Section 6, and plan to extend the dataset and training pipeline to support multiple valid trajectories and more flexible learning objectives in future versions.
> We appreciate the opportunity to clarify these points and will incorporate this discussion into the final version of the paper.

---

### Official Review · Reviewer_iZ2g · 2025-07-03

**Rating:** 4
**Confidence:** 3

**Summary:**

This paper focuses on addressing the challenges in automating professional CAD software, which requires complex 3D reasoning, multi-step workflows, and precise interactions, making AI-driven automation difficult due to a lack of large-scale training data. To overcome this, the paper introduces VIDEOCAD, a synthetic dataset of 41K annotated CAD task videos, enabling learning from human-like modeling sequences. This work also proposes VIDEOCADFORMER, a transformer-based model that improves long-horizon action prediction by 20%, and VIDEOCADQA, a benchmark to evaluate LLMs’ spatial reasoning in CAD tasks. This work bridges critical gaps in AI for CAD automation, providing tools for behavior cloning, UI understanding, and 3D reasoning.

**Dataset Code Accessibility:**

Yes

**Ethical Considerations:**

No, there are no or only very minor ethics concerns

**Final Justification:**

The authors addressed most of my questions. I will keep my original score.

**Limitations Weaknesses:**

1. The improvement of the proposed data over the Medium metric is relatively limited in Table 4. What is the reason to cause this?
2. The example in Figure 2 is quite simple. Will there be any 'chain-of-thoughts' in addition to the answer?

**Strengths Contributions:**

1. The basic idea of VIDEOCAD, which utilizes Rule-based UI bot from human-made CAD models into executable sequences of UI actions—enabling relatively large-scale data collection—is interesting and reasonable.

2. The data is both large-scale and richly annotated.

3. The paper is well-written and easy to understand.

---

> ### Author Rebuttal · Authors · 2025-07-30
>
> We thank the reviewer for the questions. We address the posed questions in order.
> ### **1. Stronger Evaluation Confirms VideoCADFormer’s Robustness**
>  In the original Table 4, we evaluated geometric fidelity on a random subset of 100 samples from the full 2000-sample test set. To improve statistical reliability, we evaluated an additional 100 samples (totaling 200), and the updated results are reported below:
>
> **Table: Updated performance by sequence length on Chamfer success rate (<0.02 ↑), mean Chamfer distance (↓), and invalid sample rate (↓)**
>
> | Method         | Success Short (%) ↑ | Success Medium (%) ↑ | Success Long (%) ↑ | Success Mean (%) ↑ |
> |----------------|---------------------|-----------------------|--------------------|--------------------|
> | Human Expert   | **85.0**            | **96.7**              | **82.8**           | **88.2**           |
> | Random         | 2.5                 | 0.0                   | 0.0                | 0.8                |
> | VPT            | 43.9                | 36.4                  | 5.9                | 28.5               |
> | VideoCADFormer | **60.6**            | **37.9**              | **25.0**           | **41.0**           |
>
> | Method         | Mean CD Short ↓ | Mean CD Medium ↓ | Mean CD Long ↓ | Mean CD Mean ↓ |
> |----------------|------------------|-------------------|-----------------|----------------|
> | Human Expert   | **0.0097**       | **0.0067**        | **0.0112**      | **0.0092**     |
> | Random         | 0.1038           | 0.1075            | 0.0972          | 0.1028         |
> | VPT            | 0.0260           | 0.0290            | 0.0856          | 0.0473         |
> | VideoCADFormer | **0.0238**       | **0.0286**        | **0.0592**      | **0.0374**     |
>
> ### Measured Uncertainty Validates Significance of Results
>
> To quantify uncertainty, we estimate the standard error of the success rate using:
> $$
> \sigma = \sqrt{\frac{p(1 - p)}{n}}, \quad n = 200
> $$
>
> For the Medium-length category:
>
> - **VPT:** $p = 0.364 \Rightarrow \sigma \approx 3.4\%$
> - **VideoCADFormer:** $p = 0.379 \Rightarrow \sigma \approx 3.4\%$
>
> ### VideoCADFormer Excels in Structural Fidelity and Robustness
>
> The modest difference between VPT and VideoCADFormer in Medium sequences falls within statistical uncertainty, despite increasing the sample size. However, our model shows consistently higher performance in Short and Long sequences, and achieves lower Chamfer distance and fewer invalid CAD models across all sequence lengths.
>
> We hypothesize that VPT tends to generate sequences that initially resemble the ground truth and thus performs adequately on Medium sequences with moderate complexity. However, as sequence complexity increases (i.e., Long category), it fails to preserve structure and correctness, leading to sharp drops in geometric fidelity. In contrast, VideoCADFormer maintains spatial and parametric consistency over long horizons, demonstrating better generalization and robustness.
>
> We will include these updated results and standard errors in the revised manuscript for clarity and completeness.
> ### **2. Explaining the simplicity of Figure 2**
>
> Thank you for your comment. Figure 2 was designed to illustrate the core construction paradigm used in the dataset (i.e., sketch–extrude–repeat), serving as a conceptual overview of how human-authored parametric CAD models are decomposed into step-by-step design operations. The simplicity of this example was intentional, to clearly convey the foundational sketch-extrude logic and corresponding UI action mapping used throughout VideoCAD to readers without a CAD background. We will add complex examples of CAD geometry that our dataset contains.
> ### **3. Explaining Chain of Thought in the data**
> Regarding the question of "chain-of-thoughts": if the reviewer is referring to intermediate reasoning steps or decision-making traces, we clarify that VideoCAD does not explicitly annotate verbal or symbolic reasoning trajectories (e.g., "why" a shape was constructed in a certain way). However, each data sample does include a full sequence of timestamped UI actions, intermediate UI frames, and high-level CAD operations, which together reflect the procedural logic and evolution of the design—offering a form of implicit chain-of-thought grounded in user actions and visual transitions.
> Moreover, since the dataset provides granular action traces and corresponding visual state changes, it is straightforward for a downstream method to reconstruct or model explicit chain-of-thought reasoning from these inputs. In other words, while the dataset does not contain verbal rationales, it contains all the necessary ingredients for generating or learning structured reasoning processes.

---

### Decision · Program_Chairs · 2025-09-18

**Decision:**

Accept (poster)

**Comment:**

This paper introduces VideoCAD, a large-scale synthetic dataset comprising annotated videos of UI interactions in CAD software, designed to address the challenges of long-horizon, precision tasks in professional tools. All reviewers acknowledged the dataset's novelty, scale, and value for complex UI interaction research. While key concerns were raised regarding the illustration of complex examples and restricted CAD operation coverage, the authors' rebuttal effectively addressed these points through additional analyses that strengthened their results. The Area Chair agrees with the reviewers' positive assessment and recommends acceptance of this paper.

===== FINAL UPDATE FROM DB Track PCs ====

The final decision for this paper has been taken by the program chairs after consultation with the SACs. All Senior Area Chairs have ranked papers according to the feedback from the AC during the review process. We decided to leave the original meta-review to reflect the opinion of the AC in light of the initial discussions with reviewers and SAC.